# ● Analysis of extinction properties as a function of relative humidity using a κ-EC-Mie model in Nanjing

Zefeng. Zhang[1], Yan. Shen[2], Yanwei. Li[1], Bin. Zhu[1], Xingna. Yu[1]

[1]Collaborative Innovation Center on Forecast and Evaluation of Meteorological Disasters, Key Laboratory for Aerosol-Cloud-Precipitation of China Meteorological Administration, Nanjing University of Information Science & Technology, Nanjing, 210044, China

[2]Fenghua Meteorological Bureau of Zhejiang Province, Fenghua 315500, China

*Correspondence to*: Y. L Li (liyw@nuist.edu.cn)

**Abstract** The relationship between relative humidity (RH) and extinction properties is of widespread concern. In this study, a hygroscopic parameter (κ) and the volume fraction of elemental carbon (EC) were used to characterize the chemical characteristics of particles, and a core-shell model was built based on these characteristics. The size distribution, chemical composition and RH were measured in Nanjing from 15/10/2013 to 13/11/2013. The model-derived extinction coefficients of particles were fitted with the program of coated spheres according to Bohren & Huffman (BHCOAT), and modeled the values correlated well with the measurement-derived extinction coefficients ($r^2$=0.81), which suggested that the core-shell model produced reasonable results. The results show that more than 81% of the extinction coefficient in Nanjing was due to particles in the 0.2-1.0μm size range. Under dry conditions, the higher mass fraction of particles in the 0.2-1.0μm size range caused the higher extinction coefficient. An increase in RH led to a significant increase in the extinction coefficient, although the increases differed among the different size segments. For $\lambda$=550 nm, the extinction coefficient from the 0.01-0.2μm, 0.2-0.5μm, and 1.0-2.0μm size ranges increased significantly with the increase in RH, whereas the extinction contributions from the 0.5-1.0μm and 2.0-10.0μm size ranges to the extinction coefficient decreased slightly.

## 1 Introduction

Degradation of visibility is likely the most readily perceived impact of aerosol pollution and it has been used as a visual indicator of ambient air quality (Watson, 2002). Visibility throughout the world has generally decreased in recent decades, especially in Asia. In China, horizontal visibility has significantly decreased since 1980 (Che et al., 2007; Qian and Giorgi, 2000; Qian et al., 2007; Streets et al., 2008; Fu

et al., 2013; Yang et al., 2015; Liu et al., 2015). For example, in Guangzhou, one of the largest cities in the Pearl River Delta (PRD) , low visibility occurs on 150 days per year (Deng et al., 2008). In the Beijing-Tianjin-Hebei region, the annual average number of continuous haze events has increased, accounting for more than half of the total haze days in a year(Zhang et al., 2015). In the Yangtze River

Delta (YRD) region, visibility has decreased at the rate of 2.4 km•decade$^{-1}$(Gao et al., 2011). Visibility degradation is mainly caused by the increase of particle number or mass concentration. The increase of particulate pollution can lead to a variety of health problems (such as cardiovascular disease, respiratory system diseases, etc.) and can further lead to an increase in traffic accidents, which has a serious impacts on human health and activities(Tie et al., 2009; Wu et al., 2005; Chang et al., 2009). As a result,the

visibility problems have received unprecedented attention in recent years.

Under dry conditions, the extinction of particles is the main factor affecting visibility (Covert et al., 1972; Deng et al., 2008; Watson, 2002). The particle number size distribution, chemical composition and relative refractive index are the important parameters that affect the optical properties of the particles(Day et al., 2000; Ma et al., 2012; Cheng et al., 2008a; Wen and Yeh, 2010). However, many of

the aerosol components are hygroscopic and take up water as a function of the relative humidity (RH) (Clarke et al., 2004; Covert et al., 1972; Swietlicki et al., 2008). When the RH is high, even at subsaturated conditions, the hygroscopic growth of the particles can lead to an increase in size and a decrease in the refractive index, which has significant effects on the extinction properties(Cheng et al., 2008b; Covert et al., 1972; Stock et al., 2011). Furthermore, the physicochemical properties of aerosols

can lead to variable hygroscopic growth, and the extinction associated with different particles (with differences in size and chemistry) is significantly different under the same RH. Overall, visibility will decrease when the RH increases (Charlson, 1969; Covert et al., 1972; Stock et al., 2011; Day and Malm, 2001). Some studies show that extinction can increase by more than 100% when the RH exceeds 70%-80% (Mcmurry, 2000; Zhang and Mcmurry, 1992; Tang, 1996). Therefore, the study of the effect of RH on

the extinction coefficient is very important.

Interest in the relationship between aerosol composition, RH and visibility dates back to at least to the studies of Wright (1940) on the atmospheric opacity over Valentia, Ireland (Wright, 1940). Currently, we can calculate the extinction coefficient accurately based onMie theory (Bohren and Huffman, 2008) as long as   we have information of the overall aerosol population. However, the atmospheric particles is a

complicated mixture in terms of chemical composition, and it is very difficult to obtain complete data on

the physicochemical properties of all the particles. The RH dependency of the extinction coefficients from atmospheric observations can be characterized by fitting a certain formula to the observations. However,this method cannot reflect the differences in physicochemical properties of particles based on the observed aerosol, and fitting curves are different at different sites(Chen et al., 2010; Yu et al., 2016).

Another possibility is to use an empirical formula which is based observed on chemical composition and different RH to calculate the extinction coefficient. However, this empirical formula may not be suitable for other locations. Therefore, we have established a model that has few variables and for which each variable can be obtained based on conventional observations. Our modeling approach consists of three components.(Cheng et al., 2006). The first component is elemental carbon (EC), which is the

light-absorbing component. The real and imaginary parts of EC are extremely high, and a typical value is 1.8- 0.54$i$ (Lee and Tien, 1981; Redemann et al., 2000). Water is the second component, which only scatters the incident radiation with a lowest refractive index of 1.33-0.0$i$ (Levoni et al., 1997). Aside from EC and water, the rest of the aerosol components primarily only scatter light, and their refractive indices are very similar, with a real part of approximately 1.53 and an imaginary part of nearly zero. This

component is called a non-light-absorbing component (Lin et al., 2013; Tang, 1996; Wex et al., 2002). Wex's study showed that the simplification is reasonable (2002). They found out that, under dry conditions, there was no statistically significant effect on the deviation between the measured and calculated scattering coefficients, when varying the mass fractions of the nearly pure light-scattering compositions within their general concentration levels. Thus, as long as we know the volume fraction of

EC and the total volume concentation, we can rather well describe the extinction properties of the particles. However, the Particles show hygroscopic growth as the RH increases. Petters and Kreidenweis (2007) proposed a simple hygroscopic parameter, κ, that can be used to calculate the hygroscopic growth factor (GF) at different levels of RH. κ can be considered a function of the volume fraction of the hygroscopic components which are non-light absorbing and the volume fraction of the non-hygroscopic

components which are light-absorbing and can be assumed to be EC here. In this way hygroscopic and optical properties can be understood to have a strong linkage. Supposing that non-light-absorbing material is uniformly mixed with water after hygroscopic growth, we can determine the changes in volume of both the real part and imaginary parts of the particles, and then we can calculate the extinction coefficient of the particles (Chen et al., 2012).Therefore, we can calculate the extinction properties of the

particles accurately at different levels of RH based on the Mie theory according to the three-component

model, as long as we obtain the total volume concentration,volume fraction of EC of the observed aerosols, the hygroscopic parameter ($\kappa$) of the observed aerosols and the hypothesized mixed mode of the observed aerosols.

In this study, the volume fraction of EC and the hygroscopic parameter ($\kappa$) were obtained using film sampling. The film sampling was used to collect data on chemical composition of particles.  In this study, the volume fraction of EC was obtained from the film sampling conducted using an  Anderson instrument in Nanjing and $\kappa$ was calculated according to the Zdanovskii-Stokes-Robinson(ZSR) rule(Petters and Kreidenweis, 2007; Stokes and Robinson, 1966). The extinction coefficient calculated by the above method had a good relationship with the visibility, which verified the reasonableness of our method. Based on this result, we further explored the growth curve of the extinction coefficient as a function of of RH and probed into the contribution of different size ranges to the extinction coefficient.

## 2 Experiment and methods

### 2.1 Measurement location and sampling

The sampling site was on the roof of a twelve-story building at the Nanjing University of Information Science and Technology in the Pukou District of Nanjing (32.207 °N; 118.717 °E), 40 m above ground level. Visibility and meteorological parameters were obtained from a meteorological station near the sampling site at a distance of less than 1.5 km. The sampling period was from 15/10/2013 to 13/11/2013. The periods of instrumental observation are shown in Fig. 1, and the missing data were due to power failure. Moreover, because the aerosol measurements  in high RH is prone to errors, we excluded the data with an RH >90% and visibility <1 km.

Fig. 1 Data coverage during the observation period

### 2.2 Instruments and data analysis

### 2.2.1 Instruments

A wide-range particle spectrometer (WPS; MSP Corporation model 1000XP) is a recently introduced commercial instrument with the unique ability to measure the size distributions of aerosols with diameters from 0.01 to 10μm (Liu, 2010). The WPS combines the principles of differential mobility analysis (DMA), condensation particle counting (CPC) and laser light scattering (LPS). DMA and CPC

are used to measure particles' number concentration in the size range of 10-500nm, and LPS is used to measure particles' number concentration in the range of 0.35-10μm. One complete scan of the entire size range with a 3s scanning period for each channel takes approximately 5 min. A detailed description is provided by(An et al., 2015; Kang et al., 2013).

Anderson is a nine-stage impact sampler that is produced by the Thermo Electron Corporation ( USA). It was used to collect aerosol samples. The sampling flow rate is 28.3 L•min$^{-1}$. The size distributions are provided in section as follows: ≤0.43μm, 0.43-0.65μm, 0.65-1.1μm, 1.1-2.1μm, 2.1-3.3μm, 3.3-4.7μm, 4.7-5.8μm, 5.8-9.0μm and >9.0μm. We used cellulose filters for ionic species and quartz filters for EC and OC(organic carbon). Before use, the quartz heated were fired for 5h at 800 ℃ to lower the blank

levels of EC and OC. All of the filters were kept in a refrigerator for cryopreservation. Every sample was collected continuously for 23h and then kept in a refrigerator before analysis (Zou et al., 2014).

    Water-soluble ions were measured with a chromatograph (Metrohm model 850 professional IC, Switzerland). $Na^+$, $NH_4^+$, $Ca^{2+}$, $Mg^{2+}$, $K^+$, $F^-$, $Cl^-$, $NO_2^-$, $NO_3^-$ and $SO_4^{2-}$ were analyzed in this study. The chromatograph includes the use of a column oven, a conductivity detector, an 858 auto-injector and a

MagIC net chromatography workstation (Metrohm, Switzerland). The column oven consists of a Metrosep C4150/4.0 separation column and Metrosep A Supp 5150/4.0 separation column. The eluent was set at 3.2mmolL$^{-1}$Na$_2$CO$_3$+1.0mmolL$^{-1}$NaHCO$_3$ for anions and 1.7mmolL$^{-1}$HNO$_3$+0.7mmolL$^{-1}$ pyridine carboxylic acid for cations. The column temperature was maintained at 30 ℃. The flow-rate was 1.0mL•min$^{-1}$, and the inject volume was 20μ L. The detection limits for $Na^+$, $NH_4^+$, $K^+$, $Mg^{2+}$, $Ca^{2+}$, $F^-$,

$Cl^-$, $NO_2^-$, $NO_3^-$ and $SO_4^{2-}$ were 0.001, 0.005, 0.001, 0.001, 0.001, 0.01, 0.01, 0.01, 0.01, 0.01mg•L$^{-1}$ respectively (An et al., 2015).

    The EC and OC concentrations were determined with a thermal/optical carbon analyzer (Model 2001A, DRI). The samples were heated to 140, 280, 480 and 580 ℃ in pure He to determine OC1, OC2, OC3 and OC4, respectively. Then the samples were heated to 580, 740 and 840 ℃ in 2%O$_2$/98%He to determine

EC1, EC2 and EC3, respectively. During the heating process, some volatilized organic compounds were converted to carbon dioxide ($CO_2$) through an oxidizer (heated manganese dioxide, $MnO_2$). $CO_2$ was reduced to methane ($CH_4$) through a methanator. Finally, the $CH_4$ equivalents were quantified with a flame ionization detector (FID). The charring effect can transform part of organic carbon into pyrolysis carbon under anaerobic heating. Hence, the correction for pyrolysis was made by continuously

monitoring the filter through a 633nm He-Ne laser in order not to underestimate OC or include some

pyrolyzed OC in the EC fraction. By monitoring the change of reflected light in the heating process, the initial reflected light is an diacritical point of OC and EC (Miao et al., 2015; Zou et al., 2014).

$PM_{2.5}$ was detected with a β-ray particulate continuous monitor (Thermo Fisher) with the working principle of measuring the particles' mass concentration through the β-ray attenuation. Visibility data were collected with a CJY-1 visibility meter (CAMA Measurement & Control Equipments Co., Ltd). The visibility meter was used to measure the scattering coefficient of the particles. It has a light source with a wavelength of 940nm. The accuracy was ±10% , and the data update rate was 1min. A detailed description of these two instruments was provided previously (Yu et al., 2016).

**2.2.2 Calculation of the hygroscopic parameter (κ)**

κ can be calculated according to many methods (Liu et al., 2014; Miao et al., 2015; Petters and Kreidenweis, 2007). In this study, κ was calculated with the ZSR rule(Petters and Kreidenweis, 2007)according to the chemical composition of the particles. For an inorganic component, we considered a system containing $H^+$, $NH_4^+$, $HSO_4^-$, $SO_4^{2-}$, and $NO_3^{2-}$. We used the ion pairing method from Gysel et al. (2007), and his method is as accurate as the ADDEM model   (Topping et al., 2005). For each species, the molecular weight, κ and density are described in detail in Table 1 (Gysel et al., 2007; Kreidenweis et al., 2008; Petters and Kreidenweis, 2008; Topping et al., 2005). Moreover, we considered the effect of water-soluble organic components (WSOC) on hygroscopic growth and assumed $\kappa_{org}$=0.1 (Jimenez et al., 2009; King et al., 2010).

We obtained the mass of each pure species according to the pairing method. Supposing a dry particle's density of 1.7 g•$cm^{-3}$ (Wehner et al., 2008), we calculated the volume of the dry particles. Aside from the WSOC and the four types of inorganic components in Table 1, we assumed that other components do not contribute to the hygroscopic properties of the aerosols. According to the ZSR rule, κ is then given by Eq. (1):

$$\kappa = \sum_{i=1}^{N} \kappa_i \frac{v_{i,dry}}{V_{tol,dry}} \tag{1}$$

where N is the number of pure materials, $\kappa_i$ is the hygroscopic parameter of the $i^{th}$ pure material, $v_{i,dry}$ is the volume of the $i^{th}$ pure material in the dry condition, and $v_{tol,\,dry}$ is the total volume of the dry particle.

Table 1 Properties of each pure material component

### 2.2.3 Calculation of the hygroscopic growth factor (GF)

The hygroscopic growth behavior of particles can be described by the theory of Köhler (1936). The theory of Köhler considered the Kelvin effect and Raoult effect and established a relationship among the saturation ratio S (at sub-saturation, S is equivalent to RH), diameter and solute properties. Introducing the hygroscopic parameter κ (Petters and Kreidenweis, 2007), the hygroscopic growth factor (GF) is determined as follows in Eq. (2):

$$GF = (1 + \frac{\kappa \cdot S}{\exp\left(\frac{4\sigma_{s/\alpha} M_w}{RT\rho_w}\right) - S})^{\frac{1}{3}} \qquad (2)$$

where $M_w$ is the molecular weight of water, R is the ideal gas constant, $\rho_w$ is the density of water, T is the temperature with a value of 20 ℃, and $\sigma_{s/\alpha}$ is assumed to be the surface tension coefficient between water and air (when T=20°C, $\sigma_{s/\alpha}$=0.0728 N m$^{-1}$).

### 2.3 Methods

### 2.3.1 Model-derived extinction coefficients

Under dry conditions, the volume fraction of EC determines the volume fraction of the light-absorbing component and κ can describe the hygroscopicity of the particle. These two parameters can be used to calculate the extinction coefficient of a single particle accurately at different RH levels. The physicochemical properties can be different for the same size of particles. It is unrealistic to describe the physicochemical properties of the aerosols individually. Therefore, the internally mixed model, externally mixed model, and core-shell model are often used to describe the chemical composition of aerosol particles in practical studies (Lesins et al., 2002; Cheng et al., 2006; Hao et al., 2010). The calculated results of the extinction coefficient by the core-shell model are usually between those of the internally mixed model and externally mixed model(Hao et al., 2010). Therefore, the core-shell model was used in this study.

In this study, the core-shell model operates under the following assumptions: 1) particles of the same size have the same physicochemical properties, and particles are spherical; 2) under dry conditions, particles are composed of a light-absorbing component (EC, 1.8-0.54*i*) and a non-light-absorbing component (1.53-0*i*), and the EC is a spherical "core" that is always at the center of the particle; 3) GF is a function of κ and the hygroscopic uptake of EC is minor, and the non-light-absorbing material is uniformly mixed with water after hygroscopic exposure. Considering that the methods of film sampling and WPS differ

significantly in time resolution, we made the following assumptions: 1) the chemical compositions of particles were unchanged for a given diameter segment of Anderson sampler; 2) the chemical composition of particles remained unchanged over the course of a day.

According to the hypothesis of the core-shell model, we can calculate the model-derived extinction coefficients of the particles using Eq. (3). N was the number of size segments 67 of the WPS (0.01-10 μm). In this study, the particles in the range of 0.5-10μm were measured by LPS.

$$b_{ext,\text{model-derived}} = \sum_{i=1}^{N} Q_{ext} \times \pi(r \times GF)_i^2 \times n(r_i) \tag{3}$$

The term n represents the number concentration of size segment $N_i$ (i from 1 to 67), and $r_i$ is the median radius corresponding to $N_i$. $Q_{ext}$ is an efficiency factor calculated with the BHCOAT program, $Q_{ext}$ is defined as the extinction cross section of particle divided by the geometric cross section of particle. The input/output parameters of $Q_{ext}$ and the formulas are listed in Table 2. In Table 2, X is a scale parameter. $D_0$ is the diameter of a single particle under dry conditions. $\lambda$ is the incident light wavelength($\lambda$=550 nm, 940 nm). The wavelength of light source of the visibility meter was 940nm, and the calculated value of $\lambda$ =940nm was used to contrast with the observed value of the visibility meter. GF is the hygroscopic growth factor, which was calculated using Eq. (2). If RH is close to 0, then GF=1. The complex refractive index was calculated with the volume weighting method after the hygroscopic growth of the particle (Lesins et al., 2002).

Table 2 Input/output parameters of the efficiency factor (Q)

**2.3.2 Measurement-derived extinction coefficients**

The meteorological optical range(MOR) is determined as (Zhang, 2007):

$$\text{MOR} = \frac{1}{\sigma}\ln\frac{|c|}{\epsilon} = \frac{1}{\sigma}\ln\frac{1}{0.05} = \frac{3.0}{\sigma} \tag{4}$$

where $\sigma$ is the extinction coefficient of the particles, $\epsilon$ is the visual threshold with a value of 0.05(MOR is equal to the visibility when $\epsilon$ =0.05), and c is the target characteristic coefficient. When the target is black, c=1.

Hence, the measured extinction coefficient can be calculated from the visibility as:

$$b_{ext,\text{measurement-derived}} = \frac{1}{\text{visibility}}\ln\frac{1}{0.05} = \frac{3.0}{\text{visibiliy}} \tag{5}$$

## 3 Results and discussion

### 3.1 Aerosol properties and visibility during the measurement period

Time series of RH, visibility, measurement extinction coefficient, and $PM_{2.5}$ during the observation period is shown in Fig. 2. The measurement extinction coefficient was calculated as 3.0/visibility (Seinfeld and Pandis, 2012). The picture shows that the visibility has a strong negative correlation with $PM_{2.5}$ and RH ($r$=-0.7 and -0.62, respectively). A time series of number size distribution for dry particles is given by Fig. 3, We find that the periods with a high number concentration had a good consistency with the periods of a high $PM_{2.5}$ mass concentration ($r$=0.7) . Fig. 4 shows the time series of κ for different particle size segments. κ was calculated according to the ZSR rule, which is described in detail in Section 2.2.2. Fig. 5 shows the time series of the volume fraction of EC in different size segments, and the volume fraction of EC was calculated using data from the Anderson instrument. Fig. 4 and Fig. 5 show that κ and the volume fraction of EC changed over time, but the variation between size segments is higher compared to the variation over time within one size segment especially interactive for κ. The reason for this difference may be that the particle size was closely related to the sources.

Fig. 2 Time series of RH, visibility, extinction coefficient, and $PM_{2.5}$ during the observation period

Fig. 3 Time series of particle number size distribution (dry particles) during the observation period

Fig. 4 Time series of κ in different size segments during the observation period

Fig. 5 Time series of the volume fraction of EC for different size segments during the observation period

### 3.2 Comparative analysis of the model-derived and measurement-derived extinction coefficients by the core-shell model

Figure 6 shows the relative values of the model and measurement values of the extinction coefficient from the core-shell model. When $\lambda$=940 nm, the calculated and measured values of extinction coefficient were in good agreement ($r^2$=0.81), which indicated that using the hygroscopic parameter (κ) and volume fraction of EC to characterize the chemical characteristics of particles was reasonable. When $\lambda$=550 nm, the correlation coefficient of the calculated and measured values ($r^2$=0.714) was slightly lower compared to $\lambda$= 940 nm, mainly because that 940 nm is similar with the light source wavelength of visibility meter."

Comparing the extinction values of 550 nm and 940 nm, we found that the model-derived extinction coefficient at 550 nm was higher, mainly due to the differences in scale parameters, which led to a Q that

was larger when $\lambda$=550 nm. Because 550 nm is the most sensitive wavelength for the human eye, the following section focuses on the measurements and calculations at $\lambda$=550 nm for discussion.

Fig. 6 Relationships among the calculated and measured values based on the core-shell model ($\lambda$=550 nm, 940 nm)

**3.3 Contributing fraction of the extinction coefficient for different size segments under dry conditions**

In the core-shell model, we defined GF=1 and then used Eq. (3) to calculate the extinction coefficients of particles under dry conditions. We can calculate the extinction coefficients of the particles in different size segments with different median radii ($r$). In this study, particle size was divided into five segments: 0.01-0.2μm, 0.2-0.5μm, 0.5-1.0μm, 1.0-2.0μm, and 2.0-10.0μm. Fig. 7(a) shows the time series of different size segments to the dry aerosol extinction coefficient, and Fig. 7(b) shows the relative contribution of different size segments to the dry aerosol extinction coefficient. Fig. 7(b) shows that the relative contribution of different size fractions were significantly different. On average, the 0.2-0.5 μm and 0.5-1.0 μm ranges together contributed more than 81% of the extinction coefficients, much higher than their total $PM_{10}$ mass fraction (45%). This result suggests that, an increase in the proportion of the particles in the 0.2-1.0 μm size range in $PM_{10}$ will result in an even greater increase in the extinction capacity relative to the unit mass of the particles. This result is consistent with the results of Kang et al. (2013). To verify this point, we present Fig.8, which reflects the extinction capacity relative to the unit mass in different size segments under dry/wet conditions. The y-axis is the ratio of the extinction coefficient to the mass concentration for the different size segments. From the picture, we can find that the extinction capacity relative to the unit mass in the 0.2-2 μm range was much stronger than that of the other segments. This result explains why the particles in the 0.2-2 μm range are the most important for the reduction of the visibility, especially those in the 0.5-1 μm range.

Fig. 7 Time series (a) and the relative contributing fraction(b) of different size segments to the dry aerosol extinction coefficient

Fig. 8 Extinction capacity relative to unit mass in different size segments under dry/wet condition

Fig. 9 Time series (a) and the relative contributing fraction(b) of different size segments to the wet aerosol extinction coefficient

**3.4 Effects of relative humidity on the extinction coefficient**

For ambient RH, we can calculate the extinction coefficients of the particles in different size segments using Eq. (3). Fig. 9(a) shows the time series of different size segments to the wet aerosol extinction coefficient, and Fig. 9(b) shows the relative contributing fraction of different size segments to the wet aerosol extinction coefficient. Comparing Fig. 7 and Fig. 9, we found that the extinction coefficients of different size segments to the wet condition were larger than for the particles under dry conditions. Simultaneously, the relative contributing fraction of different size segments to the aerosol extinction coefficient underwent significant changes. Generally speaking, when particles were in the 0.01-0.2μm, 0.2-0.5μm and 1.0-2.0μm size ranges, the relative contribution fraction of the extinction coefficients all increased, especially for fine particles (Table 3). When particles were in the 0.5-1.0μm and 2.0-10.0μm size ranges, the relative contribution fraction of the extinction coefficients decreased.

Table 3 Contribution fraction of the model-derived extinction coefficients at dry/wet condition and mass fraction in $PM_{10}$ at dry condition

The growth multiples of the extinction coefficients in the different size segments (as shown in fig.10a) was calculated through wet aerosol extinction coefficients in the corresponding size segment (as shown in fig. 9a) divided by the dry aerosol extinction coefficients in the size segment (as shown in fig. 7a). The y-axis represents growth multiples of the extinction coefficients in comparison to the value at dry conditions. The x-axis represents the variability of RH. There are five fitting curves in Fig. 10(a), representing the different size segments, and the correlation coefficient ($r^2$) of each fitting curve was larger than 0.9. This result suggests that, on different days, the changes in the enhancement of extinction with the RH in the same size segment were consistent. In addition, the extinction coefficient of the particles in the 0.01-0.2μm size range increased the most with the increased RH, followed by the extinction coefficients of particles in the 0.2-0.5μm and 1.0-2.0μm size ranges. The extinction coefficients of particles in the 0.5-1.0μm and 2.0-10.0μm size ranges did not increase with the increased RH.

The impact of RH on the particles was reflected in two aspects: the variability in diameter and the efficiency factor (Q). The growth of particles was determined by the hygroscopic parameter (κ). As κ increased, GF also increased. Fig. 4 shows the time series of κ for different particle sizes during the observations. The particles in the 0.5-1.0μm range had the largest κ, which means that the variability in

the diameter cannot explain the lack of obvious increase in the extinction coefficients in the 0.5-1.0μm

size range. To obtain Q following the influence of RH, we performed the following calculation. Firstly,

we assumed that the RH had no effect on Q, which means that Q was equivalent to the value under dry

conditions. Secondly, we calculated the extinction coefficient of the particles in the different size

segments using Eq. (3) (indicated by the letter $b_{ext}$). Extinction coefficients shown in fig.9(a) was divided

by $b_{ext}$ to produce Fig.10(b), which represented the variation in Q with respect to RH. Fig. 10(b) shows

that Q increased significantly in the 0.01-0.2μm, 0.2-0.5μm, and 1.0-2.0μm size ranges with the increase

in RH and that Q declined slightly in the 0.5-1.0μm and 2.0-10.0μm size ranges at high RH values. The

fitting curve and the calculated values are significantly different, which reflects how the change of scale

parameters coincides with the change of efficiency factors Q. For particles in the 0.01-0.2 μm, 0.2-0.5

μm, 0.5-1.0 μm and 1.0-2.0 μm size ranges, the correlation coefficients are all high. For particles in the

2.0-10.0 μm range, the correlation coefficient is very low. This finding suggested that the effect of the

variation in the scale parameter on Q was significantly different on different days.

Because the average particle size distribution and chemical composition in each size segment are known,

we can calculate the average contribution fraction of the extinction coefficients in each size segment with

the increase in RH. The calculation results are shown in Fig. 11, which illustrates that the extinction

coefficient was primarily related to particles in the 0.2-0.5μm and 0.5-1.0μm size ranges. Generally

speaking, an increase in RH will lead to an increase in the extinction coefficient, but the rate of increase

in the extinction coefficient was significantly different in each size segment. With an increase in RH, the

fractions of the extinction coefficients contributed by the 0.01-0.2μm, 0.2-0.5μm, and 1.0-2.0μm size

ranges increased considerably, whereas the fractions of the extinction coefficient contributed by the

0.5-1.0μm and 2.0-10.0μm size ranges decreased slightly.

Fig. 10 Growth multiples of the extinction coefficients (a) and the change in the efficiency factor (b)

for different size segments at ambient relative humidity

Fig. 11 Relationship between the contribution fraction of the extinction coefficient in different size

segments and relative humidity (RH)

## 4 Conclusions

In this study, a hygroscopic parameter (κ) and the volume fraction of elemental carbon (EC) were used to

characterize the chemical characteristics of particles and a core-shell model was built based on these characteristics. In the core-shell model, the real part and the imaginary part of the refractive index, the scale parameters were both functions of RH. The extinction coefficients of particles fitted with the BHCOAT program correlated well with the measured values ($r^2$=0.81) that were derived from the visibility, which suggested that using κ and the volume fraction of EC to characterize the chemical characteristics of particles was reasonable.

In the core-shell model, when $\lambda$=550 nm, the contribution fractions of the extinction coefficient of different size segments were significantly different. Under the dry condition, more than 81% of the extinction coefficients in Nanjing were contributed by particles in the 0.2-1.0μm size range, a much higher percentage than their $PM_{10}$ mass fraction (45%). This finding suggested that, for $PM_{10}$, an increase in the mass proportion of particles in 0.2-1.0μm size range results in an even greater increase in the extinction capacity.

With the increase in RH, the extinction capacity of the particles will grow significantly. In this study, the formula for the increase in the extinction coefficients in different size segments is given. At given RH, the growth rate of extinction coefficients differs significantly among the different size segments. The growth rates are related to κ, and the variation in the scale parameter leads to variations in Q, which is the main reason that the growth multiples of the extinction coefficient differ at different RH values. With the increase in RH, the extinction coefficient contribution fractions increase for particles in the 0.01-0.2μm, 0.2-0.5μm and 1.0-2.0μm size ranges but decrease for particles in the 0.5-1.0μm and 2.0-10.0μm size ranges.

**Acknowledgement:**

This work was supported jointly by the National Natual Science Foundation of China (Grant No. 41275152 and 41005071) , and the National Key Research and Development Program of China (2016YFA0602003).

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

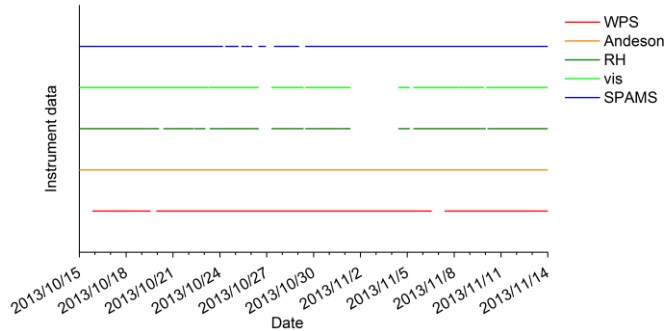

Fig. 1 Data coverage during the observation period

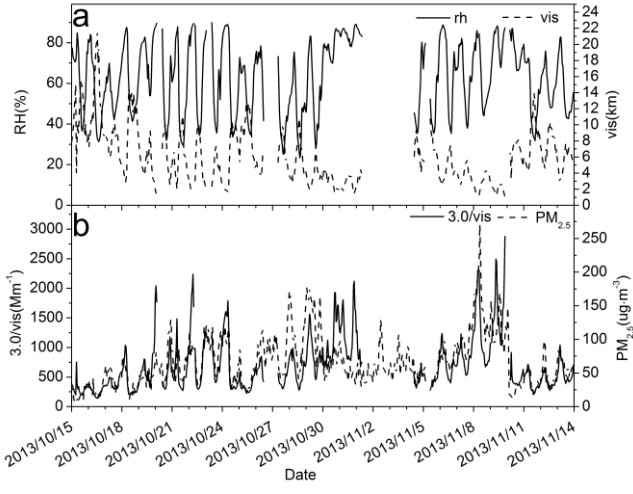

Fig. 2 Time series of RH, visibility, extinction coefficient, and PM$_{2.5}$ during the observation period

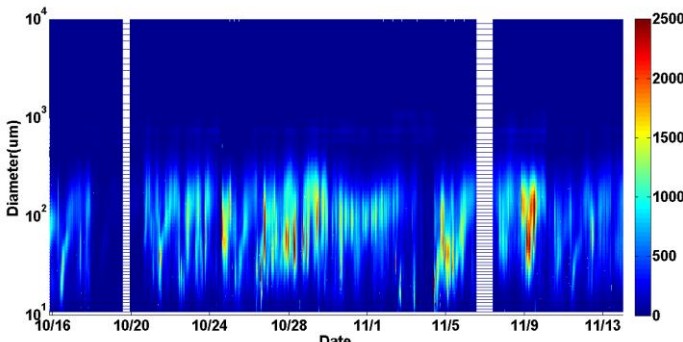

Fig. 3 Time series of number size distribution (dry particles) during the observation period

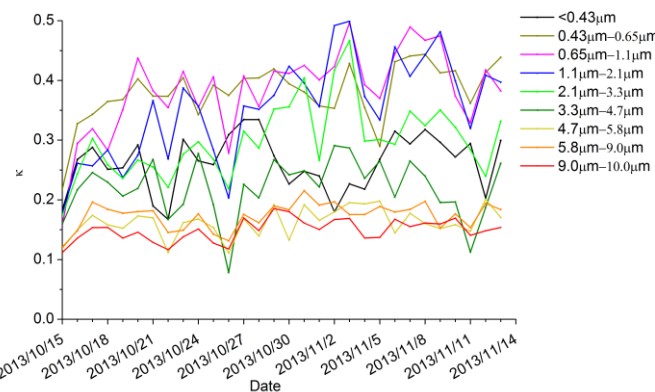

Fig. 4 Time series of κ in different size segments during the observation period

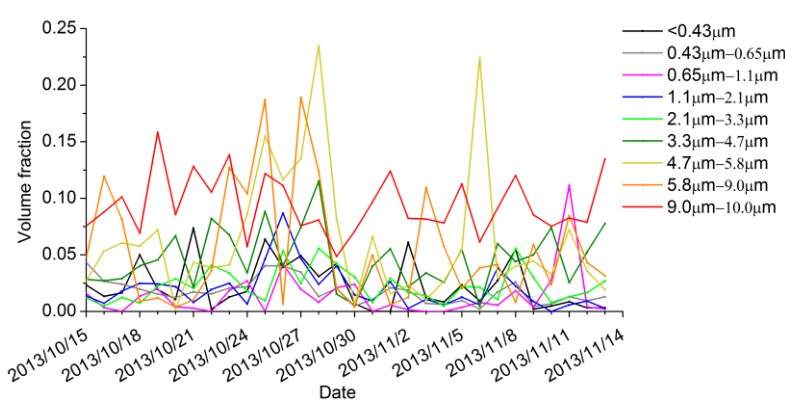

Fig. 5 Time series of the volume fraction of EC for different size segments during the observation period

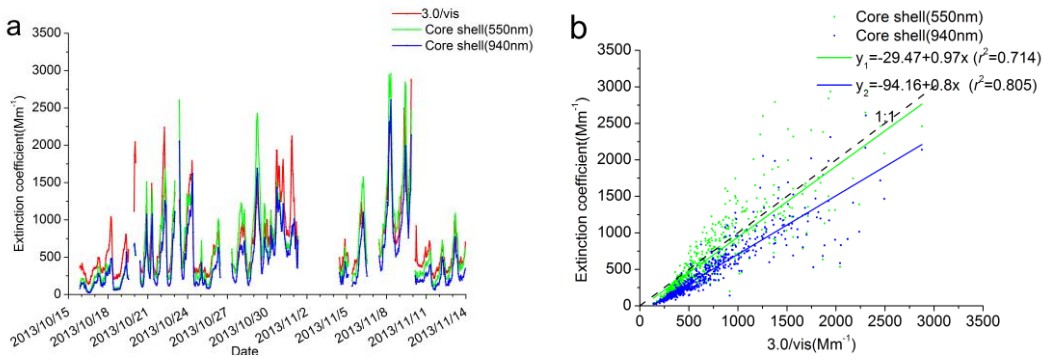

Fig. 6 Relationships among the calculated and measured values based on the core-shell model (λ=550 nm, 940 nm)

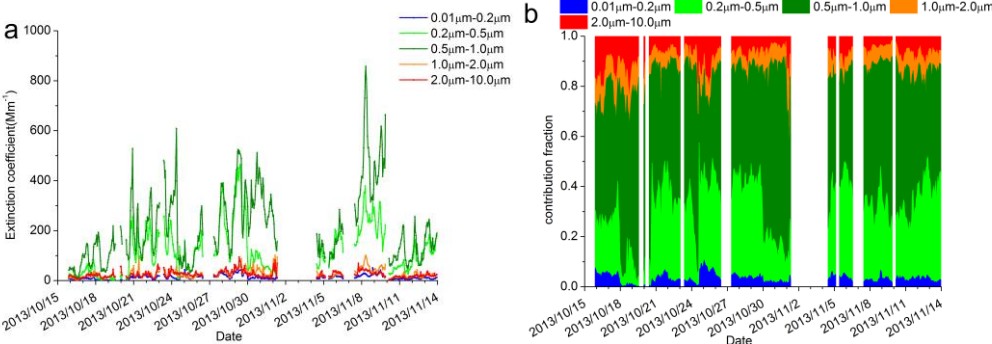

Fig. 7 Time series (a) and the relative contributing fraction(b) of different size segments to the dry aerosol extinction coefficient

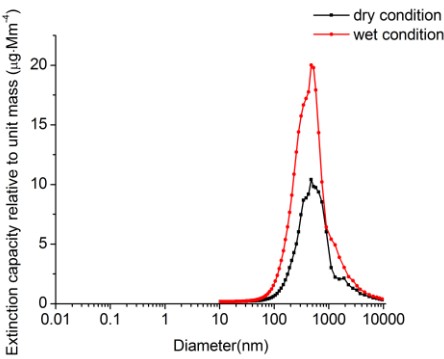

5    Fig. 8 Extinction capacity relative to unit mass in different size segments under dry/wet condition

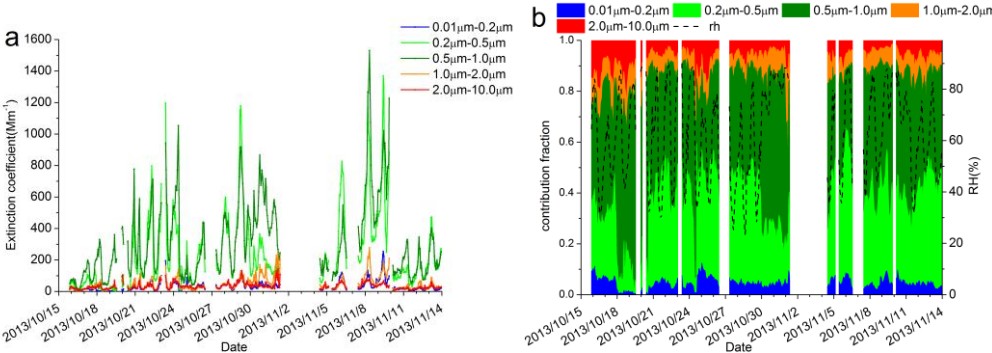

Fig. 9 Time series (a) and the relative contributing fraction(b) of different size segments to the wet aerosol extinction coefficient

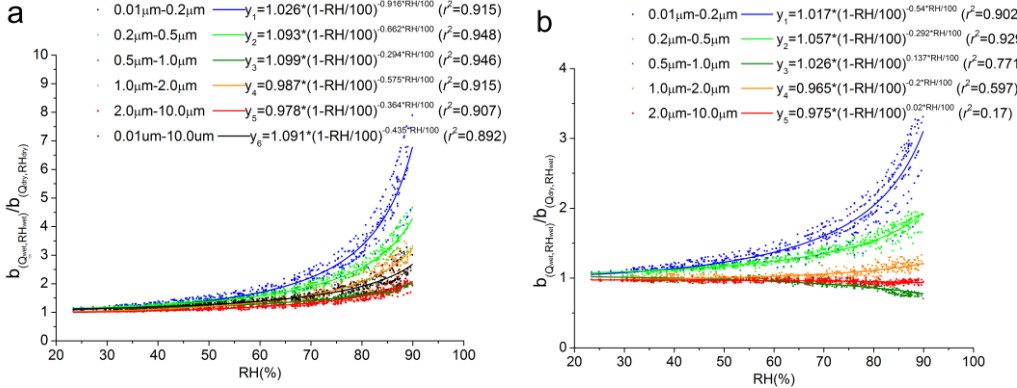

10    Fig. 10 Growth multiples of the extinction coefficients(a)and the change of efficiency factor (b) in different size segments at ambient relative humidity

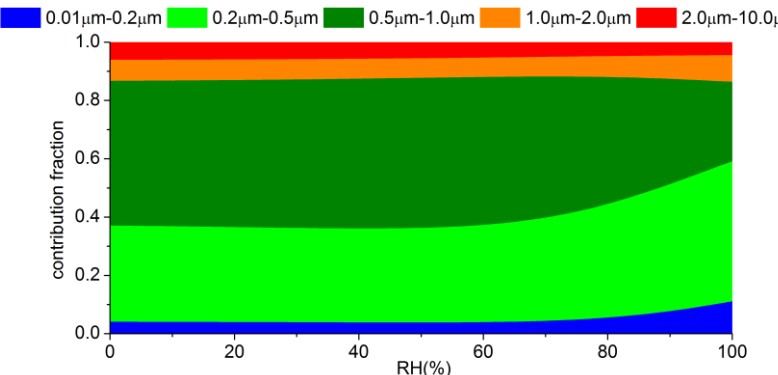

Fig. 11 Relationship between the contribution fraction of extinction coefficients in different size segments and relative humidity (RH)

5    Table 1 Properties of each pure material

|  | Molecular weight | Density (g cm$^{-3}$) | $\kappa$ |
|---|---|---|---|
| NH$_4$NO$_3$ | 80.04 | 1.72 | 0.68 |
| H$_2$SO$_4$ | 98.08 | 1.83 | 1.13 |
| NH$_4$HSO$_4$ | 115.11 | 1.78 | 0.56 |
| (NH$_4$)$_2$SO$_4$ | 132.14 | 1.77 | 0.53 |
| WSOC |  | 1.40 | 0.10 |

Table 2 Input/output parameters of efficiency factor (Q)

| input parameters | output parameters |
|---|---|
| $X_{cor} = \dfrac{\pi \cdot \sqrt[3]{\frac{V_{EC}}{V_{tol}}} \times D_0}{\lambda}$ | |
| | $Q_{ext}$ |
| $X_{man} = \dfrac{\pi \cdot D_0 \cdot GF}{\lambda}$ | $\omega_0$ |
| $m_{cor} = (1.8, 0.54)$ | |
| $m_{man} = \left(\dfrac{1.53 + 1.33(GF-1)^3}{(GF-1)^3 + 1}, 0\right)$ | |

Table 3 Contribution fraction of the model-derived extinction coefficients at dry/wet condition and mass fraction in

10    PM$_{10}$ at dry condition

|  | 0.01-0.2μm | 0.2-0.5μm | 0.5-1.0μm | 1.0-2.0μm | 2.0-10.0μm |
|---|---|---|---|---|---|
| Contribution fraction of the model-derived extinction coefficients at dry condition | 3.4% | 28.3% | 52.6% | 7.6% | 8.0% |
| Contribution fraction of the model-derived extinction coefficients at wet condition | 4.6% | 33.3% | 47.6% | 7.8% | 6.9% |
| Mass fraction in PM$_{10}$ at dry condition | 8.5% | 17.4% | 27.6% | 13.2% | 33.3% |