# Peer review of "Analysis of extinction properties as a function of relative humidity using a κ-EC-Mie model in Nanjing"

_Atmospheric Chemistry and Physics, 2016_

## Referee Comment (RC1) · Anonymous Referee #2 · 6 Jun 2016

Referee Comment

General comment:

The manuscript presents interesting results on aerosol extinction modelling using microphysical aerosol parameters obtained by other measurements. Also, the introduction gives a reasonable overview about the recently published articles in the area. A simple core-shell model is used to compare experimental and modelled data. The dataset is in general a bit short, but sufficient to test the model. Figures and tables are in most cases clear, but figure descriptions need some more work. Especially the paper text is sometimes not fluent and some English correction is recommended before publication. The discussion part is precise, but I suggest to add one or two more figures

presenting the results that were used for the model exercises. In general, the Nanjing dataset is used as in a lab study only to test the model. I suggest to put the measurements at this special place into context to similar measurements that were carried out at other sites. The article may be published in ACP when the following comments are taken into account.

Detailed scientific comments:

Abstract

Page 1, Line 15 – 22: Comment: You sometimes use extinction, then extinction coefficient, then volume extinction coefficient. Be clear and specify the parameter you are talking about!

1 Introduction

Page 2, Line 13: Add: . . . even at subsaturated conditions.

Page 2, Line 13: Change: . . .of the particles can lead . . .

Page 2, Line 16-17: . . .and the extinction associated with different particles is . . . Comment: What do you men here with different particles: size classes, chemistry? Please clarify!

Page 2, Line 25: Change: . . . of the overall aerosol population.

Page 2, Line 29: Add: . . . based on the observed aerosol.

Page 3, Line 2: Change: Therefore, we have established . . .

Page 3, Line 18: Comment: I would rather more say here that kappa can be considered a function of the volume fraction of the hygroscopic and therewith non-light absorbing components and the volume fraction of the non-hygroscopic and therewith light-absorbing component which here can be assumed to be EC. In this way hygroscopic and optical properties can be understood to have a strong linkage.

2 Experiment and Methods

Page 4, Line 18-24: Comment: Which parameter for the size distribution is measured: mass, number? Please specify!

Page 4, Line 26: Change: The size distributions are provided in section as follows: …

Page 5, Line 2: Comment: I would expect here a short description (two sentences) of the processing steps.

Page 5, Line 4: Comment: A detailed description of what was provided?

Page 5, Line 6: Comment: Again, I would expect here a short description!

Page 5, Line 9-10: Comment: Check this sentence, this is not a full sentence!

Page 5, Line 10-14: Comment: Again, I would expect here a short description of the visibility meter!

Page 5, Line 20-21: Comment: You have to specify which method you used (Gysel or Topping)?

Page 6, Line 1: Comment: Specify exactly what are the parameters i, N, etc. in formula (1).

Page 6, Line 12-14: Comment: Be correct! Change: … molecular weight of water, … the ideal gas constant, T the temperature with a value of 20 degree C, …

3 Results and Discussion

Page 6, Line 18: Comment: I would like an explanation here why this simplification of calculation of the extinction coefficient based on visibility measurements can be done. Please give here the scientific background!

Page 6, Line 20 Comment: If you use the wording size distribution, you have to specify which parameter you mean. In this case you mean particle number size distribution, I guess. Please take care over the whole manuscript! Also in Figure 3 description.
Page 6, Line 20-22: Comment: Can you give some statistical evidence on this statement. You should look for the correlation coefficient here.

Page 6, Line 25 and ongoing: Comment: I would say the variation between size segments is higher compared to the variation over time within one size segment especially for kappa. Please make sure that you mention the time resolution of the Anderson in your study in section 2! I cannot see it there.

Page 7, Line 13 – 15: Please rephrase this sentence. Physicochemical properties can be different for the same size of particles, but in this case the chemical composition is different.

Page 7, Line 27 and ongoing: Comment: This section contains some information that should be relocated to the instrumental section 2 or even is there now. Please shorten and make more clear!

Page 8, Line 12: Add: . . . of the extinction coefficient . . .

Page 8, Line 20: Change: . . . the following section focusses on the measurements and calculations at lambda = 550 nm for discussion.

Page 9, Line 1: Comment: I would rather call this the relative contributing fraction of different size segments to the dry aerosol extinction coefficient. The same I suggest for Figure 7 and Figure 8 descriptions and all other parts of the manuscript.

Page 9, Line 3 and ongoing: Comment: This is really an interesting finding. But I still understand that you measured PM2.5 and not PM10! I suggest here that a plot showing the overview of particle number or even better particle mass size distribution is included, maybe only as an average value for the whole campaign.

Page 9, Figure 9: Comment: Figure axis units and legends/symbols are too small. Figure description is incomplete as missing for Figure 9b.

Page 9, Line 13 – 22: Comment: Put the absolute measurements in context to data

from other sites you will find in the literature! Otherwise, you only use Nanjing as a lab study to test your model.

Page 9, Line 23 – Page 10, Line 11: What can you conclude from this finding. Can you look into the chemistry and find possible explanations!

Page 10, Line 12 – Page 11, Line 1: Comment: Honestly, I got lost in this paragraph. Please choose a more appropriate way to explain the details of your calculations and corresponding findings!

4 Conclusions

Page 11, Line 10: Add: . . . measured values (. . .) that were derived from . . .

Detailed technical and language comments:

Abstract

Page 1, Line 15: Change: . . . produced reasonable results.

1 Introduction

Page 2, Line 3-4: Change: . . . is mainly caused by the increase of particle number or mass concentrations and can lead. . .(. . .such as cardiovascular diseases. . .) and can further lead to an increase of traffic accidents ...

Page 2, Line 28: Change . . . though. . .

Page 3, Line 9: Change: . . . indices. . .

Page 3, Line 11: Change: She found . . .

Page 3, Line 15: Comment: There are two points after the sentence.

Page 3, Line 21: Change: . . . of the particles. . .Then, we can calculate. . .

Page 3, Line 24: Add: . . . of the observed aerosols.

Page 4, Line 3: Change: . . . in dependence of RH and . . .different particle size ranges.

2 Experiment and Methods

Page 4, Line 8: Shorten: . . . above ground level.

Page 5, Line 6: Change: EC and OC were determined . . .The measurement principle. . .

Page 5, Line 25-26: Change: We obtained . . . density of 1.7 . . . we calculated. . .

Page 6, Line 3: Add: . . .material component

3 Results and Discussion

Figure 1: Change: Data coverage . . .

Figure 2: Change: . . .observation period.

Figure 3: Change :. . . particle number size distribution . . .

Page 6, Line 21 Change: . . . that the periods with high number concentrations had a good consistency with . . .

Page 6, Line 20-22: Change: . . . for different particle size segments . . .

Figure 4: Change : . . . in different size segments ...

Page 6, Line 20-22: Change: . . . for different particle size segments . . .

Page 7, Line 18: Change: . . . the calculated results . . .

Page 7, Line 24 – 25: Change: . . . GF is a function of kappa and the hygroscopic uptake of EC is minor . . .

Page 8, Line 23: Change: Contributing . . .

Page 9, Figure 26: Change: . . . was larger than . . .

4 Conclusions

Page 11, Line 24 Change: ... size ranges.

---

## Referee Comment (RC2) · Anonymous Referee #1 · 27 Jun 2016

This contribution presents analysis of extinction properties of ambient particles in an urban environment in China (Nanjing) based on the information of particle number size distribution and particle phase chemical composition using an $\kappa$-EC-Mie model. The modeled results were compared with the measured ones obtained from the visibility data; also the relationship between the extinction properties of ambient particle and RH is discussed in the text.

Major comments:

The background information of theory and method used in this study is far too simplified, and it could be written out more explicitly in the introduction and method parts. When the author used Mie theory to calculate the extinction coefficient, more back-

ground information based on the theory need to be included into the introduction section. Similarly, the background information on the internally mixed model, externally mixed model and especially the core-shell model, which is a major method for the current paper, should be discussed more in the introduction part, for instance, the histories, the situations that the model works, the uncertainties of each model and pros and cons for each model.

The Experiment and methods part was not written in a satisfactory way. It is highlighted in the title that the extinction coefficient of the particles is the major results here, but the author did not include how you calculated or measured this parameter in the method part. Relevant paragraphs in the results section should be moved up into the method section.

Could the author reproduce some of the figures? The fonts and colors of some of the figures are too difficult to read. Please also consider presenting the figures in some other way. Time series of many parameters might not be necessary, but rather the scatter plots for the comparison of two variables that the author mentioned in the text are needed, for instance, it is really difficult to compare Fig. 7a and Fig. 8a, that I cannot fully agree with the author's statement in the text without a direct comparison of the variables from those two figures.

The author used the core-shell model to calculate the extinction coefficient, considering the particles consisting of a light-absorbing component and a non-light component. When the particles grew in size due to hydration, we ended up with a change in the geometric cross section areas of particles, which affect their scattering. Also the refractive index for scattering also changed as the particles hydrated. So please clarify how you used the core-shell model to calculate the extinction, should be summation from scattering and absorption. As well, the Q values you used in the calculation should be explained more specifically. Should the Q be the summation from scattering parameter and absorption parameter?

In Fig.2 in your manuscript, ambient RH varied from around 40% to 80%. The author calculated the size-segregated kappa based on the ZSR mixing rule. However, under low RH conditions, the ZSR mixing rule might not hold, as the water activity decreases, solute concentrations increase and the interaction between different solutes might not be neglected.

The author calculated Eq. 2 in the manuscript from Eq. 1 based on the assumptions that the aerosols were internally mixed. Then the obtained GF was used to calculate the extinction coefficient as in Fig. 6b to verify the core-shell model is reasonable. Did the author try the internally-mixed/externally-mixed model and how is the correlation coefficient between the calculated and measured extinction coefficient. Were they much worse than the current values? It is dangerous to conclude that the core-shell model was reasonable, when the author tested only one model.

Minor comments:

Page 1, line 13: Please give the full name for an abbreviation, which was mentioned for the first time in the article.

Page 1, line 27-28: Please recheck the references and the literatures for the last 2-3 years could be added.

Page 2, line 4: Do you mean particle number concentration or mass concentration, please clarify, also in Page 2, line 9.

Page 2, line 16-17: Please rephrase the sentence, as the information given is not clear.

Page 2, line 25: Avoid using words like 'if' or 'of course (page 3, line 1, line 14)', or 'oldest' (Page 3, line 27). Tell the facts about the results; avoid using too many oral statements. Please check this through the whole text.

Page 2, line 28: please change 'thought' to 'though'.

Page 3, line 2: What kind of uncertainties? From where, please clarify.

Page 3, line 5: I guess you mean 'The real and imaginary parts of the refractive index for EC'. EC cannot have imaginary part.

Page 3, line 8: similar to what? With water or EC, please clarify.

Page 3, line 11: Please give a little bit more details about Wex's study.

Page 3, line 15: I think some particles did not show hygroscopic growth when RH increases from 20% to 30%. Please recheck your statement.

Page 3, line 27-28: Please consider changing the sentence.

Page 3, line 30: The Zdanovskii-Stokes-Robinson (ZSR) mixing rule (Zdanovskii, 1948; Stokes and Robinson, 1966) is not correctly cited here.

Page 4, line 12: of which measurement error is high at high RH. And what kind of error, please specify.

Page 4, line 25: please check the full name of the instrument.

Page 4, line 26: Change 'the size distributions' to 'the ranges of each size distribution'. What is the time resolution of your impact sampler measurement?

Page 5, line 22: Please rephrase the sentence.

Page 6, line 1: For Eq. 1, please give proper explanation of each variable.

Page 6, line 17: delete 'a'. It is 'Time series of . . .'.

Page 6, line 18: Should the author put this sentence 'the extinction coefficient was calculated. . .' to method part. Also, please specify it more clearly, as you also have had other method to calculate this parameter, avoid confusing. For instance, you can define this one as measurement-derived extinction coefficient and the other one as model-derived one. Also, should the extinction coefficient was calculated as 3.9/visibility. Please check the equation.

Page 6, line 19: The author wrote that Fig. 2 shows that the visibility has a strong negative correlation with PM2.5 and RH and also gave the correlation coefficient. Please consider add scatter plot of these three variables.

Page 6, line 21: For 'had a good consistent with periods of high PM2.5', please consider either highlighting the periods or making a scatter plot.

Page 6, line 25: I would like to see in which particle size range the particles mass dominated.

Page 6, line 27: How could the particle size relate to the sources? Please specify, and which sources?

Page 7, line 1-3: Please specify how these assumptions relate to the different time resolution of film sampling and WPS measurements.

Page 7, line 10-20: Put this section into the method part.

Page 7, line 18: What kind of calculation results?

Page 7, line 21-26: In this article, you have light-absorbing component as well as non-light absorbing component. But you used visibility meter to measure scattering coefficient. Please check your visibility data. Then you ended up with the comparison of extinction from different method. Moreover, when the particles undergo hygroscopic growth, the scattering from the non-light-absorbing component was be enhanced. Please specify all of these relationships more explicitly. Clarify what the author really wants to tell.

Page 8, line 1: Give proper reference for Q, and explain it more specifically. Give the full name for BHCOAT.

Page 8, line 6: Please rephrase the sentence: its calculated value was consistent with human eye. How could the scientific results consistent with human eye?

Page 8, line 7: For 'If RH=0, GF=1', I guess you want to say under dry conditions, the particles did not take up water, thus GF as 1 was used as the input in Eq. 3 for

calculating the extinction coefficient of particles under dry conditions.

Page 8, line 12: Please specify which values.

Page 8, line 13: How could the time series be in good agreement, please rephrase the sentence.

Page 8, line 19: You cannot compare the results when different wavelength was used. Only compare the one using wavelength as 940 nm and specify that for the analysis on chemical composition and RH dependency, wavelength of 550 nm was used.

Page 8, 13-19: When you talk about something in good agreement, not only the correlation coefficient, but also the fitting line and its deviation from 1 to 1 line should be discussed.

Page 8, line 19: What are scale parameters, where did you define them?

Page 8, line 27-28: I guess you mean you fitted the measured particle number size distribution into five segments. Please clarify.

Page 9, line 3-5: Please rephrase the sentence 'On average….'. The information is not clear. And when did you measure PM10 mass fraction?

Page 9, line 4-6: You mean an increase in the mass concentration or number concentration? What do you mean 'unit mass' here, please consider changing the sentence.

Page 9, line 7: The results of Kang et al., (2013) are from a different season, please clarify.

Page 9, any results/figures on chemical composition in particle phase should be given. Page 9, line 20: It is not obvious to see the contribution fraction of the extinction coefficients from the mentioned size ranges increased, as well as the decreasing trend. Consider presenting it in another way.

Page 9, line 23: Which results, please clarify.

[Figure]

Page 9, line 27: It is actually not the extinction coefficient any more, but the extinction enhancement. Please be careful with your statement.

Page 9, line 12-30, please consider rewriting this paragraph. The information is not clearly given in the text. You cannot say divide a figure with another figure to get a third figure.

Page 10, Conclusions: This is more like a summary but not a conclusion. When you only tested the core-shell model, you cannot say it is reasonable without the information from other model results. And where did you get the PM10 mass fraction information. Where does your 45% come from?

Please also note the supplement to this comment:
http://www.atmos-chem-phys-discuss.net/acp-2016-119/acp-2016-119-RC2-supplement.pdf

---

## Author Comment (AC1) · 18 Aug 2016

**Referee #1**

This contribution presents analysis of extinction properties of ambient particles in an urban environment in China (Nanjing) based on the information of particle number size distribution and particle phase chemical composition using an к -EC-Mie model. The modeled results were compared with the measured ones obtained from the visibility data; also the relationship between the extinction properties of ambient particle and RH is discussed in the text.

**Major comments:**

**Q1**: The background information of theory and method used in this study is far too simplified, and it could be written out more explicitly in the introduction and method parts. When the author used Mie theory to calculate the extinction coefficient, more back-ground information based on the theory need to be included into the introduction section. Similarly, the background information on the internally mixed model, externally mixed model and especially the core-shell model, which is a major method for the current paper, should be discussed more in the introduction part, for instance, the histories, the situations that the model works, the uncertainties of each model and pros and cons for each model.

**A1**: First, in the method part, we added section2.3, which includes some information from section3.2 as well as the calculation of the measurement-derived extinction coefficients, on Page7, Line11- Page8, Line27. Second, we analyzed the results of the internally/externally mixed model and found them to be consistent with the core-shell model. The values of the extinction coefficient calculated by the core-shell model are between those of the internally mixed model and the externally mixed model. Considering the article length and structure, we only discussed the core-shell model in this paper. The input/output parameters of the efficiency factor (Q) and the calculation of the model-derived extinction coefficients from the internally/externally mixed model are as follows:

Table1 Input/output parameters of efficiency factor (Q) by internally mixed model

| input parameters | output parameters |
|---|---|
| $X_{\text{internally}} = \dfrac{\pi \cdot D_0 \cdot GF}{\lambda}$ | |
| | $Q_{ext}$ |
| $m_{\text{internally}} = (\dfrac{\frac{V_{EC}}{V_{tol}} \times 1.8 + \frac{V_{other}}{V_{tol}} \times 1.53 + 1.33(GF-1)^3}{(GF-1)^3 + 1}, \dfrac{\frac{V_{EC}}{V_{tol}} \times 0.54}{(GF-1)^3 + 1})$ | |

$$b_{ext,internally} = \sum_{i=1}^{N} Q_{ext} \times \pi(r \times GF)_i^2 \times n(r_i)$$

Table2 Input/output parameters of efficiency factor (Q) by externally mixed model

| input parameters | output parameters |
|---|---|
| $X_{\text{EC}} = \dfrac{\pi \cdot D_0}{\lambda}$ | |
| $X_{\text{other}} = \dfrac{\pi \cdot D_0 \cdot GF'}{\lambda}$ | $Q_{abs,EC}$ |
| $m_{\text{EC}} = (1.8, 0.54)$ | |

| | $Q_{sca,other}$ |
|---|---|
| $m_{\text{other}} = (\dfrac{1.53 + 1.33(GF' - 1)^3}{(GF' - 1)^3 + 1}, 0)$ | |

$$b_{ext,externally} = \sum_{i=1}^{N} Q_{abs,i} \times \pi \cdot r_i^2 \times n(r_i) \times (\frac{V_{EC}}{V_{tol}})_i + \sum_{i=1}^{N} Q_{sca,i} \times \pi \cdot (r \cdot GF')_i^2 \times n(r_i) \times (1 - \frac{V_{EC}}{V_{tol}})_i$$

**Q2**:The Experiment and methods part was not written in a satisfactory way. It is highlighted in the title that the extinction coefficient of the particles is the major results here, but the author did not include how you calculated or measured this parameter in the method part. Relevant paragraphs in the results section should be moved up into the method section.

**A2**: In the method part, we added section2.3, which includes the calculation of the model-derived extinction coefficients and the measurement-derived extinction coefficients. The relevant paragraphs in the results section have been moved up to section2.3.1.(on Page7, Line11-Page8,Line27)

**Q3**:Could the author reproduce some of the figures? The fonts and colors of some of the figures are too difficult to read. Please also consider presenting the figures in some other way. Time series of many parameters might not be necessary, but rather the scatter plots for the comparison of two variables that the author mentioned in the text are needed, for instance, it is really difficult to compare Fig. 7a and Fig. 8a, that I can not fully agree with the author's statement in the text without a direct comparison of the variables from those two figures.

**A3**: We have enlarged the fonts of all figures in our paper. However, we do not grasp what the reviewer means about the time series of many parameters not being necessary. Could the reviewer point out which parameters are not necessary? The time series of the various parameters are either observation results or data that the core-shell model needs. We consider these parameters to be useful. On Page11 Line5-6, 'Comparing Fig. 7 and Fig. 8, we found that the extinction coefficients of different size segments to the wet aerosol extinction coefficient were larger than for particles under dry conditions'clearly should refer to Fig. 7a and Fig. 8a, which is the foundation for the following figures.

**Q4**:The author used the core-shell model to calculate the extinction coefficient, considering the particles consisting of a light-absorbing component and a non-light component. When the particles grew in size due to hydration, we ended up with a change in the geometric cross section areas of particles, which affect their scattering. Also the refractive index for scattering also changed as the particles hydrated. So please clarify how you used the core-shell model to calculate the extinction, should be summation from scattering and absorption. As well, the Q values you used in the calculation should be explained more specifically. Should the Q be the summation from scattering parameter and absorption parameter?

**A4**: We used formula (3) to calculate the extinction coefficients. Section 2.3.1 provides a detailed introduction to this calculation (Page7, Line12– Page8, Line19).The extinction is the summation of the scattering and absorption. $Q_{ext}$ is defined as the extinction cross-section of a particle divided by its geometric cross-section' was added to Page8, Lines 8-9. $Q_{ext}$ was calculated with the BHCOAT program. The input/output parameters of $Q_{ext}$ and the formulas are listed in Table 2. $Q_{ext}$ is the summation of the scattering parameter and absorption parameter.

$$Q_{ext} = \frac{\sigma_{ext}}{\pi \cdot r^2}$$

**Q5**:In Fig.2 in your manuscript, ambient RH varied from around 40% to 80%. The author calculated the size-segregated kappa based on the ZSR mixing rule. However, under low RH conditions, the ZSR mixing rule might not hold, as the water activity decreases, solute concentrations increase and the interaction between different solutes might not be neglected.

**A5**: We agree with the reviewer's statement that under low-RH conditions, the ZSR mixing rule might not hold. Under low-RH conditions, there is no deliquescence and therefore no hygroscopic growth. Not only the ZSR rule, but all of the parameterization schemes of hygroscopic growth have the same problem. However, no reference explicitly gives a minimum RH below which the ZSR rule might not hold. The reasons may be as follows. First, the weathering point is lower than the deliquescence point, especially in the mixed state. There is no explicit weathering point, and hygroscopic growth may occur at low RH. Second, the GF is small at low RH. For example, GF=1.044 and 1.1 when RH=10%, 20%, respectively( κ =0.4). The extinction coefficient under low RH is not very different from the extinction coefficient in the dry condition. The effect of RH mainly appears at high RH.

**Q6**: The author calculated Eq. 2 in the manuscript from Eq. 1 based on the assumptions that the aerosols were internally mixed. Then the obtained GF was used to calculate the extinction coefficient as in Fig. 6b to verify the core-shell model is reasonable. Did the author try the internally-mixed/externally-mixed model and how is the correlation coefficient between the calculated and measured extinction coefficient. Were they much worse than the current values? It is dangerous to conclude that the core-shell model was reasonable, when the author tested only one model.

**A6**: We also analyzed the results of the internally/externally mixed model and found them to be consistent with those of the core-shell model. The values of the extinction coefficient calculated by the core-shell model are between those of the internally mixed model and externally mixed model. Considering the article length and structure, we discussed only the core-shell model in this paper. The relationships between the model-derived extinction coefficients and measurement-derived extinction coefficients using the three mixed models are given below ( λ =550nm).The result of the internally /externally mixed model is consistent with the current values.

[Figure]

Fig.1 Time series of model-derived extinction coefficients (core-shell/internally/externally model) and measurement-derived extinction coefficients (a); Correlation coefficient of model-derived extinction coefficients (core-shell/internally/externally model) and measurement-derived extinction coefficients (b)

**Minor comments:**

**Q7**: Page 1, Line 13: Please give the full name for an abbreviation, which was mentioned for the first time in the article.

**A7**: Thank you for reviewer's comment. We have modified the 'BHCOAT program' to 'program of coated spheres form Bohren& Huffman (BHCOAT)' on Page1 Line 15

**Q8**: Page 1, Line 27-28: Please recheck the references and the literatures for the last 2-3years could be added.

**A8**: We reviewed several related papers according to the reviewer's suggestion and added two papers from the last 2-3 years to Page2, Line1.

Liu, X., Hui, Y., Yin, Z. Y., Wang, Z., Xie, X., and Fang, J.: Deteriorating haze situation and the severe

haze episode during December 18–25 of 2013 in Xi'an, China, the worst event on record,

Theoretical & Applied Climatology, 1-15, 2015.

Yang, Y. Q., Wang, J. Z., Gong, S. L., Zhang, X. Y., Wang, H., Wang, Y. Q., Wang, J., Li, D., and Guo,

J. P.: PLAM - a meteorological pollution index for air quality and its applications in fog-haze

forecasts in North China, Atmospheric Chemistry & Physics, 16, 9077-9106, 2015.

**Q9**:Page 2, Line 4: Do you mean particle number concentration or mass concentration, please clarify, also in Page 2, Line 9.

**A9**: We have clarified the expression 'particle concentration' to' particle number or mass concentration' on Page2, Line6 and 'particle size distribution' to' particle number size distribution' on Page2, Line 12.

**Q10**:Page 2, Line 16-17: Please rephrase the sentence, as the information given is not clear.

**A10**: We have modified 'Furthermore, the physicochemical properties of aerosols lead to variable hygroscopic growth, and the extinction associated with different particles is significantly different under the same RH.' to 'Furthermore, the physicochemical properties of aerosols can lead to variable hygroscopic growth, and the extinction associated with different particles (with differences in size and chemistry) is significantly different under the same RH. ' on Page2 Lines 19-21.

**Q11**:"Page 2, Line 25: Avoid using words like 'if' or 'of course (Page 3, Line 1, Line 14)', or' oldest' (Page 3, Line 27). Tell the facts about the results; avoid using too many oral statements. Please check this through the whole text.

**A11**:'if' was revised to 'as long as' on Page2, Line29 andPage3Line18. 'Of course' was revised to 'However' on Page3, Line6.

**Q12**: Page 2, Line 28: please change 'thought' to 'though'.

**A12**: We have changed 'thought' to 'though' on Page3,Line2

**Q13**: Page 3, Line 2: What kind of uncertainties? From where, please clarify.

**A13**: We have modified the 'this method leads to great uncertainty' to 'this empirical formula may not be suitable for other locations' on Page3 Line6.

**Q14**:Page 3, Line 5: I guess you mean 'The real and imaginary parts of the refractive index for EC'. EC cannot have imaginary part.

**A14**: EC has an imaginary part, as noted in the previous literature. For example,

EC /soot (1.97-0.65i) in Hasan and Dzubay(1983)

EC /soot (1.9-0.55i) in Sloane(1984)

EC /soot (1.75-0.4i) in Redemann(2000)

Hasan H, Dzubay TG. Apportioning light extinction coefficients to chemical species in atmospheric aerosol[J]. Atmospheric Environment. 1983, 17(8): 1573-1581.

Sloane CS. Optical properties of aerosols of mixed composition[J]. Atmospheric Environment. 1984, 18(18): 871-878.

Redemann J, Turco R, Liou K et al. Retrieving the vertical structure of the effective aerosol complex index of refraction from a combination of aerosol in situ and remote sensing measurements during TARFOX[J]. Journal of Geophysical Research: Atmospheres (1984–2012). 2000, 105(D8): 9949-9970.

**Q15**:Page 3, Line 8: similar to what? With water or EC, please clarify.

**A15**: Similar to each other not water or EC. We have clarified'…with very similar refractive indexes ' to '…and their refractive indices were very similar,…'on Page3 Line13.

**Q16**: Page 3, Line 11: Please give a little bit more details about Wex's study.

**A16**:'She found that, under dry conditions, there was no statistically significant effect on the deviation between the measured and calculated scattering coefficients when varying the mass fractions of the nearly pure light-scattering compositions within their general concentration levels' on Page3 Line16-19 refers to Wex's study.

**Q17**:Page 3, Line 15: I think some particles did not show hygroscopic growth when RHincreases from 20% to 30%. Please recheck your statement.

**A17**:In the sentence 'Particles show hygroscopic growth as the RH increases' on Page3, Line 20, we describe the general trend of the particles' hygroscopic growth.

**Q18**: Page 3, Line 27-28: Please consider changing the sentence.

**A18**: We delete this sentence because we do not think it has a proper meaning.

**Q19**: Page 3, Line 30: The Zdanovskii-Stokes-Robinson (ZSR) mixing rule (Zdanovskii, 1948;Stokes and Robinson, 1966) is not correctly cited here.

**A19**: Thank you for providing this reference. We cited'(Petters and Kreidenweis, 2007)' here because in this paper, the ZSR rule was linked up with κ after the establishment of the Köhler equation. However, we also added the original source that the reviewer provided here on Page4, Line8. '…κ was calculated according to the ZSR rule(Petters and Kreidenweis, 2007; Stokes and Robinson, 1966). '

**Q20**:Page 4, Line 12: of which measurement error is high at high RH. And what kind of error, please specify.

**A20**: We modified '…measurement error is high at high RH' to '…system measurement error is high at high RH' on Page4, Line20. The measurement of the RH is not accurate at high RH. The system measurement error is approximately 2% at low RH, while it could exceed 5% when the RH is higher

than 90%. Small perturbations in the temperature will also affect the measurement of the RH when the RH is high. Because the RH is such an important parameter in our study, we need to exclude the data for which RH >90%.

**Q21**:Page 4, Line 25: please check the full name of the instrument.
**A21**: The full name of the instrument is Anderson. We have verified this.

**Q22**:Page 4, Line 26: Change 'the size distributions' to 'the ranges of each size distribution'. What is the time resolution of your impact sampler measurement?
**A22**:We have modified the sentence expression according to Referee #2 Q10.'The size distributions were' to 'The size distributions are provided in section as follows ' on Page5,Lines7-8. We have added the Anderson's time resolution on Page5 Line12.'Every sample was collected continuously for 23h and then kept in a refrigerator before analyzing '

**Q23**: Page 5, Line 22: Please rephrase the sentence.
**A23**: We rephrase the sentence into ' Moreover, we considered the hygroscopic effect of water-soluble organic components (WSOC) and assumed $\kappa_{org}$=0.1 (Jimenez et al., 2009; King et al., 2010).'

**Q24**: Page 6, Line 1: For Eq. 1, please give proper explanation of each variable.
**A24**:We explained the parameters in formula (1) 'where N is the number of pure materials, $\kappa_i$ is the hygroscopic parameter of the $i^{th}$ pure material, $v_{i,dry}$ is the volume of the $i^{th}$ pure material in the dry condition, and $v_{tol,\ dry}$ is the total volume of the dry particle.'

**Q25**:Page 6, Line 17: delete 'a'. It is 'Time series of : : :'.
**A25**: We have deleted 'a' on Page9 Line3.

**Q26**:Page 6, Line 18: Should the author put this sentence 'the extinction coefficient was calculated: : :' to method part. Also, please specify it more clearly, as you also have had other method to calculate this parameter, avoid confusing. For instance, you can define this one as measurement-derived extinction coefficient and the other one as model-derived one. Also, should the extinction coefficient was calculated as 3.9/visibility. Please check the equation.
**A26**: We added section 2.3 and moved some of the information in section3.2 to section 2.3.1. The extinction coefficients have been clarified as measurement-derived and model-derived throughout the whole paper. The derivation of the term3.0/visibility was provided in section 2.3.2.

**Q27**: Page 6, Line 19: The author wrote that Fig. 2 shows that the visibility has a strong negative correlation with $PM_{2.5}$ and RH and also gave the correlation coefficient. Please consider add scatter plot of these three variables.
**A27**: In our paper, the correlation among the visibility, $PM_{2.5}$ and RH is not the main content. We therefore consider the correlation coefficient to be sufficient.

[Figure]

Fig.2 Scatter plot of visibility, PM$_{2.5}$ mass concentration, and RH

**Q28**:Page 6, Line 21: For 'had a good consistent with periods of high PM2.5', please consider either highlighting the periods or making a scatter plot.

**A28**: We added a correlation coefficient to Page9, Lines7-8. '…the periods with a high number concentration had a good consistency with the periods of a high PM$_{2.5}$ mass concentration ($r$=0.7)'

[Figure]

Fig.3 Scatter plot of WPS mass concentration and PM$_{2.5}$ mass concentration

**Q29**:Page 6, Line 25: I would like to see in which particle size range the particles mass dominated.

**A29**: We added Table3 on Page11, Lines 12-13. The particle mass dominated in the ranges of 2.0-10.0μm and 0.5-1.0μm. The mass concentrations of the particles in the different size segments were calculated by the WPS data.

Table 3 Contribution fraction of the model-derived extinction coefficients at dry/wet condition and mass fraction in PM10 at dry condition

|  | 0.01-0.2μm | 0.2-0.5μm | 0.5-1.0μm | 1.0-2.0μm | 2.0-10.0μm |
|---|---|---|---|---|---|
| Contribution fraction of the model-derived extinction coefficients at dry condition | 3.4% | 28.3% | 52.6% | 7.6% | 8.0% |
| Contribution fraction of the model-derived extinction coefficients at wet condition | 4.6% | 33.3% | 47.6% | 7.8% | 6.9% |

| Mass fraction in $PM_{10}$ at dry condition | 8.5% | 17.4% | 27.6% | 13.2% | 33.3% |
|---|---|---|---|---|---|

**Q30**:Page 6, Line 27: How could the particle size relate to the sources? Please specify, and which sources?

**A30**: Particles from different sources have different sizes. For example, particles from dust are larger than particles from biomass or traffic.

**Q31**: Page 7, Line 1-3: Please specify how these assumptions relate to the different time resolution of film sampling and WPS measurements.

**A31**: The time resolution of the WPS is 5min, and that of the Anderson is 23h. According to the hypothesis on Page7, Line28- Page8, Lines1-3, '1) the chemical compositions of particles were unchanged for a given diameter segment of Anderson' we can unify the particle size segments of the Anderson with those of the WPS. '2) the chemical composition of particles remained unchanged over the course of a day ' we can unify the time resolutions of the Anderson and the WPS to 1h.

**Q32**:Page 7, Line 10-20: Put this section into the method part.

**A32**: We added section2.3 and have moved Page 7, Lines 10-20 to this part.

**Q33**:Page 7, Line 18: What kind of calculation results?

**A33**: The results were calculated results of extinction coefficient. We have modified 'The calculated results ' to 'The calculated results of the extinction coefficient…' onPage7 Line20.

**Q34**: Page 7, Line 21-26: In this article, you have light-absorbing component as well as non-light absorbing component. But you used visibility meter to measure scattering coefficient. Please check your visibility data. Then you ended up with the comparison of extinction from different method. Moreover, when the particles undergo hygroscopic growth, the scattering from the non-light-absorbing component was be enhanced. Please specify all of these relationships more explicitly. Clarify what the author really wants to tell.

**A34**: The visibility meter measures the scattering coefficient of particles, the extinction coefficient is calculated according to the single scattering albedo obtained in the calibration of the visibility meter, and then the visibility is calculated from the extinction coefficient. The measurement-derived extinction coefficient in this paper is calculated by an empirical formula according to the visibility value that is output from the visibility meter.

**Q35**: Page 8, Line 1: Give proper reference for Q, and explain it more specifically. Give the full name for BHCOAT.

**A35**: First, 'efficiency factor ($Q_{ext}$) is defined as the extinction cross section of particle divided by the geometric cross section of particle.' was added on Page8 Line9. $Q_{ext}$ was calculated using the BHCOAT program. The input/output parameters of $Q_{ext}$ and the formulas are listed in Table 2. $Q_{ext}$ is the summation of the scattering parameter and absorption parameter.

$$Q_{\text{ext}} = \frac{\sigma_{ext}}{\pi \cdot r^2}$$

Second, we modified the 'BHCOAT program' to 'program of coated spheres form Bohren& Huffman

(BHCOAT)' on Page1 Line 15 according to Q7.

**Q36**: Page 8, Line 6: Please rephrase the sentence: its calculated value was consistent with human eye. How could the scientific results consistent with human eye?

**A36**: We have modified '550 nm is the most sensitive wavelength for the human eye, and its calculated value was consistent with the human eye.' to '550 nm is the most sensitive wavelength for the human eye, and its calculated value was consistent with the value that the most sensitive for human eye' on Page 8 Lines 13-15.

**Q37**:Page 8, Line 7: For 'If RH=0, GF=1', I guess you want to say under dry conditions, the particles did not take up water, thus GF as 1 was used as the input in Eq. 3 for calculating the extinction coefficient of particles under dry conditions.

**A37**: Yes. A GF of 1 was used as the input in Eq.3 for calculating the extinction coefficient of particles under dry conditions in this paper.

**Q38**: Page 8, Line 12: Please specify which values.

**A38**:The value here was extinction coefficient, and we have modified 'the relative values of the calculated and observed values from the core-shell model' to 'the relative values of the model and measurement values of the extinction coefficient from the core-shell model ' on Page9 Line22-23.

**Q39**: Page 8, Line 13: How could the time series be in good agreement, please rephrase the sentence.

**A39**:We have modified '…the time series of calculated and observed values were in good agreement,…' to '…the calculated and measured values of extinction coefficient were in good agreement,…' onPage9 Line23-24.

**Q40**:Page 8, Line 19: You cannot compare the results when different wavelength was used.
Only compare the one using wavelength as 940 nm and specify that for the analysison chemical composition and RH dependency, wavelength of 550 nm was used.

**A40**: In our study, the light source wavelength of the visibility meter was 940nm. We calculated the extinction coefficient of 940nm to perform a correlation between the model-derived extinction coefficients and the measurement-derived extinction coefficients. However, the output visibility value is calculated by an extinction coefficient of 550nm. Therefore, we used 550nm to calculate the absolute extinction coefficients. Because 550 nm is the most sensitive wavelength for the human eye, the following section focuses on the calculations at $\lambda$=550 nm for discussion.

**Q41**: Page 8, 13-19: When you talk about something in good agreement, not only the correlation coefficient, but also the fitting Line and its deviation from 1 to 1 Line should be discussed.

**A41**: For $\lambda$=550nm, the mean deviation of the model-derived extinction coefficients and he measurement-derived extinction coefficients is approximately 13% ([measurement-derived extinction coefficient -model-derived extinction coefficient]/ measurement-derived extinction coefficient). The measurement error of the visibility meter is approximately 10%, so we believe that the result is credible.

**Q42**: Page 8, Line 19: What are scale parameters, where did you define them?

**A42**: We mentioned the scale parameter on Page8, Line10 ('X is a scale parameter'), and we provided the formula in table 2.

**Q43**: Page 8, Line 27-28: I guess you mean you fitted the measured particle number size distribution into five segments. Please clarify.
**A43**: The data on the particle number size distribution is from the WPS. We divided it into five segments for comparison with other studies in the literature. What we fitted in the study is the extinction coefficient, not the size distribution.

**Q44**: Page 9, Line 3-5: Please rephrase the sentence 'On average: : :.'. The information is not clear. And when did you measure PM10 mass fraction?
**A44**: First, we have modified 'On average, the 0.2-0.5µm and 0.5-1.0µm ranges together contributed more than 81% of the extinction coefficients, much higher than their $PM_{10}$mass fraction (45%).' to 'On average, the 0.2-0.5µm and 0.5-1.0µm ranges together contributed more than 81% of the extinction coefficients, much higher than their total $PM_{10}$ mass fraction (45%).' on Page10 Lines 14-15. 81% and 45% can been seen in Table3.

Second, The $PM_{10}$ mass concentration was calculated from the WPS data. The mass fraction in the five size segments can be seen in Table3 (on Page11, Lines 12-13).

**Q45**: Page 9, Line 4-6: You mean an increase in the mass concentration or number concentration? What do you mean 'unit mass' here, please consider changing the sentence.
**A45**: What we mean here is an increase in the mass fraction. The phrase 'extinction capacity relative to the unit mass of the particles' is equivalent to (extinction coefficient/ mass concentration of the particles). Fig.9, which we added, shows the extinction capacity relative to a unit mass in the different size ranges.

[Figure]

Fig.4 Extinction capacity relative to unit mass in different size segments under dry/wet condition

**Q46**: Page 9, Line 7: The results of Kang et al., (2013) are from a different season, please clarify.
**A46**: The result we obtained was that the extinction capacity of the particles in the 0.2-1.0µm range was stronger than that in the other size range. This result has nothing to do with the seasons.

**Q47**: Page 9, any results/figures on chemical composition in particle phase should be given.
**A47**: We do not understand what the reviewer means here. It appears that the reviewer wants us to

emphasize the results of the specific chemical composition. In our study, we have been using κ -EC to describe the chemical composition.

**Q48**: Page 9, Line 20: It is not obvious to see the contribution fraction of the extinction coefficients from the mentioned size ranges increased, as well as the decreasing trend. Consider presenting it in another way.
**A48**: We added Table 3 on Page11 Line12-13. In this table we can see the obvious change.

Table 3 Contribution fraction of the model-derived extinction coefficients at dry/wet condition and mass fraction in PM10 at dry condition

|  | 0.01-0.2μm | 0.2-0.5μm | 0.5-1.0μm | 1.0-2.0μm | 2.0-10.0μm |
|---|---|---|---|---|---|
| contribution fraction of the model-derived extinction coefficients at dry condition | 3.4% | 28.3% | 52.6% | 7.6% | 8.0% |
| contribution fraction of the model-derived extinction coefficients at wet condition | 4.6% | 33.3% | 47.6% | 7.8% | 6.9% |
| mass fraction in $PM_{10}$ at dry condition | 8.5% | 17.4% | 27.6% | 13.2% | 33.3% |

**Q49**: Page 9, Line 23: Which results, please clarify.
**A49**: We have modified the sentence 'The results shown in Fig. 8(a) were divided by those in Fig. 7(a) to produce the results in Fig. 9(a)' to 'The calculated results of extinction coefficients shown in Fig. 8(a) was divided by the calculated results of extinction coefficients in Fig. 7(a) to produce the growth multiples results in Fig. 9(a).'on Page11 Line14-15.

**Q50**: Page 9, Line 27: It is actually not the extinction coefficient any more, but the extinction enhancement. Please be careful with your statement.
**A50**: We have modified 'extinction coefficient' to 'extinction enhancement' on Page 11, Line 19, according to the reviewer's comment

**Q51**: Page 9, Line 12-30, please consider rewriting this paragraph. The information is not clearly given in the text. You cannot say divide a figure with another figure to get a third figure.
**A51**: We have modified the sentence 'The results shown in Fig. 8(a) were divided by those in Fig. 7(a) to produce the results in Fig. 9(a)' to 'The calculated results of extinction coefficients shown in Fig. 8(a) was divided by the calculated results of extinction coefficients in Fig. 7(a) to produce the growth multiples results in Fig. 9(a).'on Page11 Line14-15.

**Q52**: Page 10, Conclusions: This is more like a summary but not a conclusion. When you only tested the core-shell model, you cannot say it is reasonable without the information from other model results. And where did you get the PM10 mass fraction information. Where does your 45% come from?
**A52**: First, the answer to Q6 applies here. We also analyzed the results of the internally/externally

mixed model and found that the result is consistent with that of the core-shell model. The values of the extinction coefficient calculated by the core-shell model are between those of the internally mixed model and externally mixed model. Considering the article length and structure, we discussed only the core-shell model in this paper. The relationships between the model-derived extinction coefficients and measurement-derived extinction coefficients by the three mixed models are given below ( $\lambda$ =550nm). The result of the internally/externally mixed model is consistent with the current values.

[Figure]

Fig.1 Time series of model-derived extinction coefficients (core-shell/internally/externally model) and measurement-derived extinction coefficients (a); Correlation coefficient of model-derived extinction coefficients (core-shell/internally/externally model) and measurement-derived extinction coefficients (b)

Second, The $PM_{10}$ mass concentration in our study was calculated from the WPS data. From table3, 45%= 17.4%+27.6%.

Please also note the supplement to this comment:

http://www.atmos-chem-phys-discuss.net/acp-2016-119/acp-2016-119-RC2-supplement.pdf

**A: The comment in the supplement is the same as that made by Referee #1, to whom we have replied.**

---

## Author Comment (AC2) · 18 Aug 2016

**Referee #2**

Referee Comment

**General comment:**
The manuscript presents interesting results on aerosol extinction modelling using microphysical aerosol parameters obtained by other measurements. Also, the introduction gives a reasonable overview about the recently published articles in the area. A simple core-shell model is used to compare experimental and modelled data. The dataset is in general a bit short, but sufficient to test the model. Figures and tables are in most cases clear, but figure descriptions need some more work. Especially the paper text is sometimes not fluent and some English correction is recommended before publication. The discussion part is precise, but I suggest to add one or two more figures presenting the results that were used for the model exercises. In general, the Nanjing dataset is used as in a lab study only to test the model. I suggest to put the measure-ACPD ments at this special place into context to similar measurements that were carried out at other sites. The article may be published in ACP when the following comments are taken into account. Interactive

**A:** First, thank you for your suggestions for this paper. We modified the content carefully according to reviewers' suggestions one by one. We apologize that there were still many sentences with grammatical errors. Thank you for your detailed revision. We have corrected these errors in the amended version and sent the manuscript to AMERICAN JOURNAL EXPERTS (AJE) for further polishing.

Second, the reviewer suggested that we put the measurements from this specific place into context based on similar measurements collected at other sites. We are looking for funding support and plan to make similar observations in Shenyang and Suzhou during the winter. We believe that we will obtain some interesting results.

**Detailed scientific comments:**
Abstract
**Q1:** Page 1, Line 15 – 22: Comment: You sometimes use extinction, then extinction coefficient, then volume extinction coefficient. Be clear and specify the parameter you are talking about!
**A1**: Thank you for your comment. We have modified 'extinction' and 'volume extinction coefficient' to 'extinction coefficient'. ( Page 1, Lines17– 23)

1 Introduction
**Q2**: Page 2, Line 13: Add:...even at subsaturated conditions.
**Q3**: Page 2, Line 13: Change:...of the particles can lead ...
**A2-A3:** We have modified the sentence expressions according to the reviewer's advice. 'When the RH is high, even at subsaturated conditions, the hygroscopic growth of the particles can lead to an increase in size…' on Page 2, Lines 16-17

**Q4**: Page 2, Line 16-17: ...and the extinction associated with different particles is ...Comment: What do you men here with different particles: size classes, chemistry? Please clarify!
**A4**: We have modified 'different particles ' to 'different particles (with differences in size and chemistry)' on Page2 Lines 20-21.

**Q5**: Page 2, Line 25: Change: ...of the overall aerosol population.

**Q6**: Page 2, Line 29: Add: ...based on the observed aerosol.

**Q7**: Page 3, Line 2: Change: Therefore, we have established ...

**A5-A7**: Thank you for reviewer's comment. We have modified the sentence expression '…on the physicochemical properties of all particles…' to '…of the overall aerosol population…' on Page 2, Line 29; 'in physicochemical properties of particles' to 'in physicochemical properties of particles based on the observed aerosol ' on Page 3, Line 3; and 'we must establish' to 'we have established ' on Page 3, Line 6.

**Q8**: Page 3, Line 18: Comment: I would rather more say here that kappa can be considered a function of the volume fraction of the hygroscopic and therewith non-light absorbing components and the volume fraction of the non-hygroscopic and therewith light-absorbing component which here can be assumed to be EC. In this way hygroscopic and optical properties can be understood to have a strong linkage.

**A8**: Thank you for your comment. The reviewer's phrasing expresses our intended meaning more clearly, so we have adopted the reviewer's expression. We modify ' $\kappa$ can be considered a function of the chemical composition and volume fraction of the non-light-absorbing component because the hygroscopic growth of EC is poor' to '$\kappa$ can be considered a function of the volume fraction of the hygroscopic and therewith non-light absorbing components and the volume fraction of the non-hygroscopic and there with light-absorbing component which here can be assumed to be EC. In this way hygroscopic and optical properties can be understood to have a strong linkage' on Page 3, Lines22-25.

2 Experiment and Methods

**Q9**: Page 4, Line 18-24: Comment: Which parameter for the size distribution is measured: mass, number? Please specify!

**A9**: The particle number concentration is used for the size distribution. We have specified as such on Page 5, Lines2-3

**Q10**: Page 4, Line 26: Change: The size distributions are provided in section as follows:...

**A10**: We have modified the sentence expression from 'The size distributions were' to 'The size distributions are provided in section as follows ' on Page5,Lines 7-8.

**Q11**: Page 5, Line 2: Comment: I would expect here a short description (two sentences) of the processing steps.

**A11**: We have added a short description 'Before use, quartz filters were fires for 5h at 800°C to lower the blank levels for EC and OC. All of these filters were kept in a refrigerator for cryopreservation. Every sample was collected continuously for 23h and then kept in a refrigerator before analyzing (Zou et al., 2014).' on Page5, Line 10-12.

**Q12**: Page 5, Line 4: Comment: A detailed description of what was provided?

**A12**: A detailed description of the chromatograph was added on Page5, Lines14-22. 'Chromatography includes the use of a column oven, a conductivity detector, an 858 auto-injector and a MagIC net chromatography workstation (Metrohm, Switzerland). The column oven consists of a Metrosep

C4150/4.0 separation column and MetrosepASupp 5150/4.0 separation column. The eluent was set at3.2mmol・$L^{-1}Na_2CO_3$+1.0mmol・$L^{-1}NaHCO_3$for anions and 1.7mmol・$L^{-1}HNO_3$+0.7mmol・$L^{-1}$ pyridine carboxylic acid forcations. The column temperature was maintained at 30°C. The flow-rate was 1.0mL・$min^{-1}$,and the inject volume was 20μL. The detection limits for $Na^+$, $NH_4^+$, $K^+$,$Mg^{2+}$,$Ca^{2+}$, $F^-$, $Cl^-$, $NO_2^-$, $NO_3^-$ and $SO_4^{2-}$ were0.001, 0.005, 0.001, 0.001, 0.001, 0.01, 0.01, 0.01, 0.01, 0.01mg・$L^{-1}$ respectively(An et al., 2015). '

**Q13**: Page 5, Line 6: Comment: Again, I would expect here a short description!
**A13**: A detailed description of the thermal/optical carbon analyzer was added on Page5, Line23-Page6 Line2. 'The samples were heated to 140, 280, 480 and 580°Cin pure He to determine OC1, OC2, OC3 and OC4, respectively. Then the samples were heated to 580, 740 and 840°Cin 2%$O_2$/98%He to determine EC1, EC2 and EC3, respectively. The volatilized compounds were converted to carbon dioxide ($CO_2$) through an oxidizer (heated manganese dioxide, $MnO_2$). $CO_2$ was reduced to methane ($CH_4$) through a methanator. Finally, the $CH_4$ equivalents were quantified with a flame ionization detector (FID). The charring effect can transform part of organic carbon into pyrolysis carbon under anaerobic heating. Hence, the correction for pyrolysis was made by continuously monitoring the filter through a 633nm He-Ne laser. By monitoring the change of reflected light in the heating process, the initial reflected light is an diacritical point of OC and EC'

**Q14**: Page 5, Line 9-10: Comment: Check this sentence, this is not a full sentence!
**A14**: We have modified '$PM_{2.5}$ was detected with a β-ray particulate continuous monitor (Thermo Fisher). And the working principles was that measuring particles' mass concentration through β-ray attenuation.' to '$PM_{2.5}$ was detected with a β-ray particulate continuous monitor (Thermo Fisher) with the working principle of measuring the particles' mass concentration through the β-ray attenuation.' on Page6 Lines 2-3.

**Q15**: Page 5, Line 10-14: Comment: Again, I would expect here a short description of the visibility meter!
**A15**: We have modified 'The visibility meter was used to measure scattering coefficient of particles and it's light source wavelength was 940nm. A detailed description of these two instruments was provided previously (Yu et al., 2015)' to ' The visibility meter was used to measure the scattering coefficient of the particles and it's light source wavelength was 940nm. The accuracy was ±10% , and the data update rate was 1min. A detailed description of these two instruments was provided previously (Yu et al., 2015) 'on Page6 Lines 6-8.

**Q16**: Page 5, Line 20-21: Comment: You have to specify which method you used (Gysel or Topping)?
**A16**: We modified the sentence 'Gysel et al. (2007) used the ion pairing method' to ' We used the ion pairing method from Gysel et al. (2007),....'Page6, Line13-14.

**Q17**: Page 6, Line 1: Comment: Specify exactly what are the parameters i, N, etc. in formula (1).
**A17**: We have more precisely specified the parameters in formula (1) on Page6, Lines 24-26. 'where N is the number of pure materials, $\kappa_i$ is the hygroscopic parameter of the $i^{th}$ pure material, $v_{i,dry}$ is the volume of the $i^{th}$ pure material in the dry condition, and $v_{tol, dry}$ is the total volume of the dry particle. '

**Q18**: Page 6, Line 12-14: Comment: Be correct! Change:...molecular weight of water, ...the ideal gas constant, T the temperature with a value of 20 degree C, ...

**A18**: We have modified the expression 'molar mass' to 'molecular weight ', 'an ideal gas constant' to 'the ideal gas constant ', and 'T=20°C ' to 'T is the temperature with a value of 20°C ' on Page7,Lines 8-9.

3 Results and Discussion

**Q19**: Page 6, Line 18: Comment: I would like an explanation here why this simplification of calculation of the extinction coefficient based on visibility measurements can be done. Please give here the scientific background!

**A19**: We added section2.3, to provide the further scientific back ground on this simplification. An explanation of 3.0/visibility was provided on Page8 Lines21-27.

'The meteorological optical range is determined as (Zhang, 2007):

$$\text{MOR} = \frac{1}{\sigma}\ln\frac{|c|}{\epsilon} = \frac{1}{\sigma}\ln\frac{1}{0.05} = \frac{3.0}{\sigma} \tag{4}$$

where $\sigma$ is the extinction coefficient of the particles, $\epsilon$ is the visual threshold with a value of 0.05(MOR is equal to the visibility when $\epsilon$ =0. 05), and c is the target characteristic coefficient. When the target is black, c=1.

Hence, the measured extinction coefficient can be calculated from the visibility as:

$$b_{\text{ext,measurement}-d} = \frac{1}{\text{visibility}}\ln\frac{1}{0.05} = \frac{3.0}{\text{visibiliy}} \tag{5}'$$

**Q20**: Page 6, Line 20 Comment: If you use the wording size distribution, you have to specify which parameter you mean. In this case you mean particle number size distribution, I Discussion paper guess. Please take care over the whole manuscript! Also in Figure 3 description.

**A20**: The size distribution here refers to the number size distribution, and we have modified 'size distribution ' to 'number size distribution ', including the description of Fig3 on Page9 Line6 and Page9 Line 16.

**Q21:** Page 6, Line 20-22: Comment: Can you give some statistical evidence on this statement. You should look for the correlation coefficient here.

**A21**: We added a reference to the correlation coefficient on Page9, Line8. '……the periods with high number concentration had a good consistency with periods ofhigh $PM_{2.5}$ mass concentration ($r$=0.7)' to '……the periods with a high number concentration had a good consistency with the periods of a high $PM_{2.5}$ mass concentration ($r$=0.7)'

[Figure]

Fig.1 Scatter plot of WPS mass concentration and $PM_{2.5}$ mass concentration

**Q22**: Page 6, Line 25 and ongoing: Comment: I would say the variation between size segments is higher compared to the variation over time within one size segment especially Interactive for kappa. Please make sure that you mention the time resolution of the Anderson in your study in section 2! I cannot see it there.

**A22**: First, we have modified the sentence 'but the degree of change in different size segments was larger than that at different times' to 'the variation between size segments is higher compared to the variation over time within one size segment especially interactive for $\kappa$' on Page9, Lines12-13. Second, we have given the time resolution of the Anderson as 23h on Page5, Line12. However, the time resolution in the whole discussion part (section 3) is 1h. Page8 Lines1-3 provids an explaination for the time resolutions of the Anderson and WPS.

**Q23**:Page 7, Line 13 – 15: Please rephrase this sentence. Physicochemical properties can be different for the same size of particles, but in this case the chemical composition is different.

**A23**: We have modified the sentence 'In the real atmosphere, even if the sizes of aerosol particles are the same, the physicochemical properties of particles are significantly different.' to 'The physicochemical properties can be different for the same size of particles, but in this case the chemical composition is different' on Page7 Lines16-17

**Q24**: Page 7, Line 27 and ongoing: Comment: This section contains some information that should be relocated to the instrumental section 2 or even is there now. Please shorten and make more clear!

**A24**: We have relocated some of the information in section 3.2 to section 2.3.

**Q25**: Page 8, Line 12: Add: ...of the extinction coefficient...

**A25**:We have modified 'Figure 6 shows the relative values of the calculated and observed values from the core-shell model' to 'Figure 6 shows the relative values of the model and measurement values of the extinction coefficient from the core-shell model 'on Page9,Lines22-23.

**Q26**: Page 8, Line 20: Change: ...the following section focusses on the measurements and calculations at lambda = 550 nm for discussion.

**A26**: We have modified the sentence 'the following section adopts 550 nm for discussion' to 'the

following section focuses on the measurements and calculations at $\lambda=550$ nm for discussion.'on Page10, Lines1-2

Q27: Page 9, Line 1: Comment: I would rather call this the relative contributing fraction of different size segments to the dry aerosol extinction coefficient. The same I suggest for Figure 7 and Figure 8 descriptions and all other parts of the manuscript.

A27: Thank you for reviewer's advice. We have modified 'of the extinction coefficients in different size segments under dry conditions' to 'of different size segments to the dry aerosol extinction coefficient' on Page 10 Line11, 'contribution fractions of the extinction coefficients in different size segments under dry conditions' to 'relative contributing fraction of different size segments to the dry aerosol extinction coefficient' on Page 10 Line11-12, 'of extinction coefficients in different size segments at ambient RH ' to 'of different size segments to the wet aerosol extinction coefficient' on Page11 Line3, 'contribution fractions of the extinction coefficients in different size segments at ambient RH' to 'relative contributing fraction of different size segments to the wet aerosol extinction coefficient ' on Page11 Line4-5. 'in different size segments at ambient RH ' to 'of different size segments to the wet condition' on Page11 Line6, 'contribution fractions of the extinction coefficients in different size segments ' to 'relative contributing fraction of different size segments to the aerosol extinction coefficient' on Page11 Line7. And the same for Figure 7, Figure 8 descriptions on Page10 Line24-27.

**Q28**:Page 9, Line 3 and ongoing: Comment: This is really an interesting finding. But I still understand that you measured PM2.5 and not PM10! I suggest here that a plot showing the overview of particle number or even better particle mass size distribution is included, maybe only as an average value for the whole campaign.

**A28**: In our study, the $PM_{10}$ mass concentration was calculated by WPS data. We added Fig.9 to Page10 Line28.'Extinction capacity relative to unit mass in different size segments under dry/wet conditions'. and a description of this figure on Page10 Lines 18-24. 'To verify this point, we present Fig.9, which reflects the extinction capacity relative to the unit mass in different size segments under dry/wet conditions. The y-axis is the ratio of the extinction coefficient to the mass concentration for different size segments. From the picture, we can find that extinction capacity relative to the unit mass in the 0.2-2 μ m range was much stronger than that of the other segments. This result explaines why the particles in the 0.2-2 μ m range are the most important for the reduction of the visibility, especially those in the 0.5-1 μ m range.'

[Figure]

Fig.2 Extinction capacity relative to unit mass in different size segments under dry/wet condition

**Q29**: Page 9, Figure 9: Comment: Figure axis units and legends/symbols are too small. Figure description is incomplete as missing for Figure 9b.

**A29**: First, because we added a picture before Fig9, the original Fig.9 has become Fig.10. We have increased the font size in Fig.10, and performed the same process on all the other pictures throughout the manuscript. We also modified the title of Figure 10a/10b on Page12 Lines 24-25. 'Growth multiples of the extinction coefficients(a)and the change in the efficiency factor (b) for different size segments at ambient relative humidity '

**Q30**:Page 9, Line 13 – 22: Comment: Put the absolute measurements in context to data from other sites you will find in the literature! Otherwise, you only use Nanjing as a lab study to test your model.

**A30**: In the literature, we usually find that the particles in the 0.2-2 μm range correlate well with the visibility. However, with the RH increasing, depth analyse of extinction in different size segments are rare. In this study, we take Nanjing as an example to perform a test and depth analysis, providing the explanation that the particles in the 0.2-2 μm range correlate well with the visibility. Now, we are looking for funding support and plan to perform observations in Shenyang and Suzhou during the winter. We believethat this study will obtain some interesting results..

**Q31**:Page 9, Line 23 – Page 10, Line 11: What can you conclude from this finding. Can you look into the chemistry and find possible explanations!

**Q31**:Fig10b provides the explanation for this part. We swapped information on Page12 Lines15-23('Because the average particle size distribution…0.5-1.0μm and 2.0-10.0μm size ranges decreased slightly.') with information on Page11 Line24-Page12 Line14 ('The impact of RH on particles was reflected…extinction coefficients vary at different RH levels')

**Q32**: Page 10, Line 12 – Page 11, Line 1: Comment: Honestly, I got lost in this paragraph. comment Please choose a more appropriate way to explain the details of your calculations and corresponding findings!

**A32**: Please see response to Q31. We swapped the information of Fig. 10b with that of Fig. 11.

4 Conclusions

**Q33**: Page 11, Line 10: Add:...measured values (...) that were derived from... Detailed technical and language comments

**A33**: We added '…that were derived from the visibility…' on Page13 Line6.

**Q34-Q58 are correction of sentences and grammatical errors. Thank you for reviewer's suggestions. We have modified these carefully, and send this paper to AMERICAN JOURNAL EXPERTS(AJE) to polish the full text again.**

Abstract

**Q34**: Page 1, Line 15: Change:...produced reasonable results.

**A34**: We have modified '…was reasonable 'to '…produced reasonable results' on Page1 Line 17.

1 Introduction

**Q35**: Page 2, Line 3-4: Change: ...is mainly caused by the increase of particle number or mass concentrations and can lead...(...such as cardiovascular diseases:::) and can further lead to an increase of traffic accidents ...

**A35**: Thank you for your advice. We have modified 'Visibility degradation mainly caused by the increase of particle concentration and easily lead to a variety of health problems (such as cardiovascular disease, respiratory system diseases, etc.) and traffic accidents increasing' to 'Visibility degradation is mainly caused by the increase of particle number or mass concentration and can lead to a variety of health problems (such as cardiovascular disease, respiratory system diseases, etc.) and can further lead to an increase of traffic accidents,' on Page2 Lines5-9.

**Q36**: Page 2, Line 28: Change ...though...

**A36**:We have modified "thought" to "though" on Page3,Line2

**Q37**: Page 3, Line 9: Change: ...indices...

**A37**: We have modified "indexes" to "indices" on Page3, Line13

**Q38**: Page 3, Line 11: Change: She found ...

**A38**: We have modified "He" to "She" on Page3, Line16

**Q39**: Page 3, Line 15: Comment: There are two points after the sentence.

**A39**: We have deleted one point here. (Page3 Line20)

**Q40**: Page 3, Line 21: Change: ...of the particles:::Then, we can calculate...

**A40**: Thank you for reviewer's advice. We have modified 'we can determine the changes in volume of both the real part and imaginary parts, then calculate the extinction coefficient of particles ' to 'we can determine the changes in volume of both the real part and imaginary parts of the particles, then we can calculate the extinction coefficient of particles ' on Page3 Lines26-28

**Q41**: Page 3, Line 24: Add: ...of the observed aerosols.

**A41**: We have added '...of the observed aerosols' on Page4 Line1.

**Q42**: Page 4, Line 3: Change: ...in dependence of RH and ...different particle size ranges.

**A42**: We have modified 'the extinction coefficient as the RH increased ' to 'the extinction coefficient in dependence of RH ' on Page4, Line11.

2 Experiment and Methods

**Q43**: Page 4, Line 8: Shorten:...above ground level.

**A43**: We have modified '40 m above the ground ' to '40 m aboveground level' on Page4 Line16.

**Q44**: Page 5, Line 6: Change: EC and OC were determined ...The measurement principle...

**A44**: We have added a detailed description of the thermal/optical carbon analyzer according, as stated in the response to Q13, on Page5, Line23-Page6 Line2. The sentence mentioned by the reviewer has been deleted.

**Q45**: Page 5, Line 25-26: Change: We obtained ...density of 1.7...we calculated...

**A45**: We have modified 'We can obtain the mass of each pure species according to the pairing method. Supposing a dry particle's density is 1.7 g•cm$^{-3}$(Wehner et al., 2008), we can calculate the volume of the dry particle.' to 'We obtained the mass of each pure species according to the pairing method. Supposing a dry particle's density of 1.7 g•cm$^{-3}$(Wehner et al., 2008), we calculated the volume of the dry particle. ' on Page6 Lines19-20.

**Q46**: Page 6, Line 3: Add:...material component

**A46**: We have added 'component' on Page6 Line27.

3 Results and Discussion

**Q47**: Figure 1: Change: Data coverage...

**A47**: We have modified 'Missing data from instruments…' to 'Data coverage from instruments…' on Page4 Line22.

**Q48**: Figure 2: Change:...observation period.

**A48**: We have added 'period' on Page20 Line6.

**Q49**: Figure 3: Change ...particle number size distribution ...

**A49**: We have modified 'Time series of size distribution (dry particles)…' to 'Time series of particle number size distribution (dry particles)…' on Page9 Line16.

**Q50**: Page 6, Line 21 Change: ...that the periods with high number concentrations had a good consistency with ...

**A50**:We have modified the sentence 'the periods of high particles number concentration had a good consistent with periods ofhigh PM$_{2.5}$ mass concentration ' to 'the periods with a high number concentration had a good consistency with the periods of a high PM$_{2.5}$ mass concentration ($r$=0.7)' on Page9 Line7-8.

**Q51**: Page 6, Line 20-22: Change: ...for different particle size segments...

**A51**: We have modified 'for different particle sizes' to 'for different particle size segments ' on Page9 Line9.

**Q52**: Figure 4: Change :...in different size segments ...

**A52**: We have modified 'Time series of κ for different sizes…' to 'Time series of κ in different size segments…' on Page9 Line17.

**Q53**: Page 6, Line 20-22: Change: ...for different particle size segments...

**A53**: We have modified 'for different particle sizes' to 'for different particle size segments ' on Page9 Line9. (same as Q51)

**Q54**: Page 7, Line 18: Change:...the calculated results...

**A54**: We have modified '…the calculation results ' to 'The calculated results...' on Page7 Line20.

**Q55**: Page 7, Line 24 – 25: Change: ...GF is a function of kappa and the hygroscopic uptake of EC is minor...

**A55**: We have modified 'GF as a function of κ, the hygroscopic level of EC is poor' to 'GF is a function of κ and the hygroscopic uptake of EC is minor…' on Page7 Lines26-27.

**Q56**: Page 8, Line 23: Change: Contributing...

A56: We have modified the sentence on Page10 Line5.

**Q57**: Page 9, Figure 26: Change: ...was larger than ::.

**A57**: We have modified '…was more than…' to '...was larger than…' on Page11 Line18.

4 Conclusions
**Q58**: Page 11, Line 24 Change:...size ranges.
**A58**: We have modified '2.0-10.0μm ranges' to '2.0-10.0μmsize ranges ' on Page13 Line21.

---

## Referee Report (RR1)

**Referee Comment**

**General comment:**

The authors have been greatly improving the manuscript according to my comments. Before final publication I have a few minor issues I would recommend to take care of. These minor comments are listed below.

**Detailed scientific comments:**

**Abstract**

Page 1, Line 13:
Comment: Which size distribution? Number? Please specify!

**1 Introduction**

Page 2, Line 6-7:
Comment: It is not the visibility that leads to health problems. Please correct the sentence!

Page 2, Line 12-14:
Comment: It is not the chemical composition itself, rather more the resulting refractive index that affect the optical properties. Please say that!

Page 2, Line 18:
… which has …

Page 4, Line 5:
Remove Q18!

**2 Experiment and Methods**

Figure 1:
Comment: I would rather more describe this as "data coverage" in the figure description.

Page 5, Line 9:
… were fired …

Page 5, Line 11:
… analysis …

Page 5, Line 22:
… EC and OC concentrations …

Page 6, Line 15:
Remove Q23!

Page 7, Line 11:
… EC determines the …

Page 8, Line 11-13:
Comment: Check the sentence "550nm is the …". It does not make sense!

**3 Results and Discussion**

Figure 4:
 … in different size segments …

Page 9, Line 26:
… lower compared to …

Figure 8:
Comment: Figure 9 is actually discussed before Figure 8. Please restructure!

Page 11, Line 14-15 and ongoing:
Comment: Check this sentence! You now talk about Figure 10! You mix up the Figure numbers in the following.

Figure 10b:
Comment: I still do not understand Figure 10b!

**4 Conclusions**

Page 13, Line 11:
… of the extinction coefficients …

---

## Author Response (AR2)

Submitted on 28 Sep 2016
Anonymous Referee #2
**Anonymous during peer-review:**  **Yes**  No
**Anonymous in acknowledgements of published article:**  **Yes**  No

Recommendation to the Editor
**1) Scientific Significance**
Does the manuscript represent a substantial contribution to scientific progress within the scope of this journal (substantial new concepts, ideas, methods, or data)?
Excellent **Good**    Fair  Poor
**2) Scientific Quality**
Are the scientific approach and applied methods valid? Are the results discussed in an appropriate and balanced way (consideration of related work, including appropriate references)?
Excellent **Good**    Fair  Poor
**3) Presentation Quality**
Are the scientific results and conclusions presented in a clear, concise, and well structured way (number and quality of figures/tables, appropriate use of English language)?
Excellent **Good**    Fair  Poor

For final publication, the manuscript should be
accepted as is
**accepted subject to technical corrections**
accepted subject to minor revisions
reconsidered after major revisions
        I would be willing to review the revised paper, if the Editor considers it necessary
        I would NOT be willing to review the revised paper
rejected

Please note that this rating only refers to this version of the manuscript!

**Suggestions for revision or reasons for rejection (will be published if the paper is accepted for final publication)**

**Referee Comment**
**General comment:**
The authors have been greatly improving the manuscript according to my comments.
Before final publication I have a few minor issues I would recommend to take care of.
These minor comments are listed below.
**Detailed scientific comments:**
Abstract
**Q1:**Page 1, Line 13:
Comment: Which size distribution? Number? Please specify!
**A1:** We have changed 'size distribution' to 'number distribution' on Page1 Line13.

**Q2:**Page 2, Line 6-7:

Comment: It is not the visibility that leads to health problems. Please correct the

sentence!

**A2:** We have clarified 'Visibility degradation is mainly caused by the increase of particle number or mass concentration and can lead to a variety of health problems (such as cardiovascular disease, respiratory system diseases, etc.)' to 'Visibility degradation is mainly caused by the increase of particle number or mass concentration. The increase of particulate pollution can lead to a variety of health problems (such as cardiovascular disease, respiratory system diseases, etc.)' on Page2 Line6-8.

**Q3:**Page 2, Line 12-14:

Comment: It is not the chemical composition itself, rather more the resulting refractive

index that affect the optical properties. Please say that!

**A3:** We have clarified 'The particle number size distribution, chemical composition and relative refractive index determined by the chemical composition are the important parameters that affect the optical properties of the particles' to 'The particle number size distribution, chemical composition and relative refractive index are the important parameters that affect the optical properties of the particles' on Page2 Line12-14.

**Q4**:Page 2, Line 18:

… which has …

**A4**: We have changed'… which have…' to '… which has…' on page2 Line18.

**Q5:**Page 4, Line 5:

Remove Q18!

**A5:** We have removed 'Q18' on Page4 Line6.

2 Experiment and Methods

**Q6**:Figure 1:

Comment: I would rather more describe this as "data coverage" in the figure description.

**A6:** We have changed 'Fig. 1 Missing data from instruments during the observation period' to 'Fig. 1 Data coverage from instruments during the observation period' on Page19 Line13.

**Q7:**Page 5, Line 9:

… were fired …

**A7:** We have changed 'were fires' to 'were fired' on Page5 Line10.

**Q8:**Page 5, Line 11:

… analysis …

**A8:** We have changed 'analyzing' to 'analysis' on Page5 Line12.

**Q9:**Page 5, Line 22:

… EC and OC concentrations …

**A9:** We have changed 'EC and OC concentration' to 'EC and OC concentrations' on Page5 Line23.

Page 6, Line 15:

**Q10:**Remove Q23!

**A10:** We have removed 'Q23' on Page6 Line16.

**Q11:**Page 7, Line 11:

… EC determines the …

**A11:** We have changed 'EC can describe the… ' to 'EC determines the …'

**Q12:**Page 8, Line 11-13:

Comment: Check the sentence "550nm is the …". It does not make sense!

**A12:** We have changed '550nm is the…' to 'Wavelength of 550 nm is the…' on Page8 Line13.

3 Results and Discussion

**Q13:**Figure 4:

… in different size segments …

**A13:** We have changed 'Fig. 4 Time series of κ in different sizes during the observation period'to 'Fig.

4 Time series of κ in different size segments during the observation period' on Page20 Line4.

**Q14**:Page 9, Line 26:

… lower compared to …

**A14:** We have changed '…lower than when $\lambda$= 940 nm' to 'lower compared to $\lambda$= 940 nm' on Page9 Line27.

**Q15:**Figure 8:

Comment: Figure 9 is actually discussed before Figure 8. Please restructure!

**A15:** We have exchanged Figure 9 and Figure 8 according to review's comment on page.

**Q16:**Page 11, Line 14-15 and ongoing:

Comment: Check this sentence! You now talk about Figure 10! You mix up the Figure
numbers in the following.

**A16:** We are so sorry that mix up the numbers. We have changed 'Fig. 9(a)' to 'Fig. 10(a)' on Page11
Line15 and ongoing.

**Q17:**Figure 10b:

Comment: I still do not understand Figure 10b!

**A17:** Fig. 10(b) represented the variation in Q with respect to RH. From Fig.10a, we find that the

extinction coefficient of particles in the 0.01-0.2μm size range increased the fastest with the increased

RH, followed by the extinction coefficients of particles in the 0.2-0.5μm and 1.0-2.0μm size ranges.

While the particles in the 0.5-1.0μm range had the largest κ, which means that the variability in diameter

cannot explain the lack of obvious increase in the extinction coefficients in the 0.5-1.0μm size range.

According to the Mie theory, the impact of RH on particles was reflected in two aspects: the variability

in diameter and the efficiency factor (Q). Thus, we give the Fig. 10(b), and find that Q increased significantly in the 0.01-0.2μm, 0.2-0.5μm, and 1.0-2.0μm size ranges with the increase in RH and that Q declined slightly in the 0.5-1.0μm and 2.0-10.0μm size ranges at high RH values. Because $\lambda$=550 nm, the increase in the scale parameter in the 0.01-0.2μm, 0.2-0.5μm, and 1.0-2.0μm size ranges favors the increase in Q, whereas the increase of the scale parameter in the 0.5-1.0μm size range leads to a decrease in Q. So, we draw the conclusion that variation in the scale parameter leads to variation in Q, which is the main reason that growth multiples of the extinction coefficients vary at different RH levels.

4 Conclusions
**Q18:**Page 13, Line 11:
… of the extinction coefficients …
**A18:** We have changed '…of the extinctions…' to 'of the extinction coefficients' on Page13 Line11

Submitted on 01 Oct 2016
Anonymous Referee #1
**Anonymous during peer-review:   Yes**  No
**Anonymous in acknowledgements of published article:       Yes**  No

Recommendation to the Editor
**1) Scientific Significance**
Does the manuscript represent a substantial contribution to scientific progress within the scope of this journal (substantial new concepts, ideas, methods, or data)?
Excellent **Good**     Fair  Poor
**2) Scientific Quality**
Are the scientific approach and applied methods valid? Are the results discussed in an appropriate and balanced way (consideration of related work, including appropriate references)?
Excellent  Good      **Fair** Poor
**3) Presentation Quality**
Are the scientific results and conclusions presented in a clear, concise, and well structured way (number and quality of figures/tables, appropriate use of English language)?
Excellent  Good       Fair **Poor**

For final publication, the manuscript should be
accepted as is
accepted subject to technical corrections
**accepted subject to minor revisions**
reconsidered after major revisions
        I would be willing to review the revised paper, if the Editor considers it necessary
        I would NOT be willing to review the revised paper
rejected

Please note that this rating only refers to this version of the manuscript!

**Suggestions for revision or reasons for rejection (will be published if the paper is accepted for final publication)**

I think the manuscript has been improved from the previous one. The results the manuscript presented are interesting and of importance, which fill the data pool of extinction properties globally. However, I still have some comments regarding to the current version:

**Q1:**

1)    The presentation or writing of the manuscript should be improved much more. There are too many long, wired, improper statements in the manuscript, for example: Page 3, line 4-6; Page 3, line 22-25; Page 4, line 5; Page 5, line 22-30; Page 7, line 14-16; Page 11, line 14-15, ect.... Please consider going through the whole manuscript and maybe asking someone who has more experience in English writing to check it.

**A1:** We amended the paper again, especially the sentences mentioned above.

**Q2:**

2)    The author presented 11 figures, eight of which were about time series. Since the article is not focusing on time series analysis, I suggest the author replace some of them as scatter plot. For instance, the author compared PM2.5 with visibility and PM2.5 with the total particle number concentration. However, it is not obvious to see how well their negative or positive correlations in the time series of Fig. 2 or Fig. 3. Hence, scatter plot might be more useful. From the time series of Fig. 4 to Fig. 8, no critical findings or conclusions were driven by the author and the information presented in these time series figures are obviously not clear. I suggest the author consider moving them into supplementary material or giving more discussion.

**A2:** We also considered what reviewer said. While the time series figures in our study are basic data, and are used to be basis of the following calculations. For instance, Fig. 2 and Fig. 3 mainly give the observed results of some variates. Fig. 4 and Fig. 5 are the key parameters in our model, and their time series are the basis of the calculation of extinction coefficients. Fig.6 to Fig.9 are the calculated results that calculated by our model.

For the correlations that reviewer mentioned, the correlation among the visibility, PM2.5 and the total particle number concentration are not the main content. We therefore only give the correlation coefficient.

**Q3:**

3)    I am not satisfied with the author's response to Question 6 from Referee 1 (see acp-2016-119-AC1-supplement-2). The internally mixed and externally mixed results are not shown in Fig. 1, which is not consistent with its figure caption and figure legend or where are the green and red curves?

**A3:** We are so sorry that referee is not satisfied with our reply to Question 6. The Fig.1 we give including the internally mixed and externally mixed results, and found them to be consistent with those of the core-shell model. Maybe the coincidence of lines caused the misunderstanding. Thus, we give them respectively. The values of the extinction coefficient calculated by the core-shell model are

between those of the internally mixed model and externally mixed model. The relationships between the model-derived extinction coefficients and measurement-derived extinction coefficients using the three mixed models are given below (λ=550nm).The result of the internally /externally mixed model is consistent with the current values. Considering the article length and structure, we discussed only the core-shell model in this paper.

Submitted on 01 Oct 2016
Anonymous Referee #1

**Anonymous during peer-review:   Yes**  No
**Anonymous in acknowledgements of published article:      Yes**  No

Recommendation to the Editor

**1) Scientific Significance**

Does the manuscript represent a substantial contribution to scientific progress within the scope of this journal (substantial new concepts, ideas, methods, or data)?

Excellent **Good**     Fair Poor

**2) Scientific Quality**

Are the scientific approach and applied methods valid? Are the results discussed in an appropriate and balanced way (consideration of related work, including appropriate references)?

Excellent  Good      **Fair** Poor

**3) Presentation Quality**

Are the scientific results and conclusions presented in a clear, concise, and well structured way (number and quality of figures/tables, appropriate use of English language)?

Excellent  Good      Fair **Poor**

For final publication, the manuscript should be

accepted as is

accepted subject to technical corrections

**accepted subject to minor revisions**

reconsidered after major revisions

       I would be willing to review the revised paper, if the Editor considers it necessary

       I would NOT be willing to review the revised paper

rejected

Please note that this rating only refers to this version of the manuscript!

**Suggestions for revision or reasons for rejection (will be published if the paper is accepted for final publication)**

I think the manuscript has been improved from the previous one. The results the manuscript presented are interesting and of importance, which fill the data pool of extinction properties globally. However, I still have some comments regarding to the current version:

**Q1:**

1)     The presentation or writing of the manuscript should be improved much more. There are too many long, wired, improper statements in the manuscript, for example: Page 3, line 4-6; Page 3, line 22-25;

Page 4, line 5; Page 5, line 22-30; Page 7, line 14-16; Page 11, line 14-15, ect.... Please consider going through the whole manuscript and maybe asking someone who has more experience in English writing to check it.

**A1:** We amended the paper again, especially the sentences mentioned above.

**Q2:**

2)  The author presented 11 figures, eight of which were about time series. Since the article is not focusing on time series analysis, I suggest the author replace some of them as scatter plot. For instance, the author compared PM2.5 with visibility and PM2.5 with the total particle number concentration. However, it is not obvious to see how well their negative or positive correlations in the time series of Fig. 2 or Fig. 3. Hence, scatter plot might be more useful. From the time series of Fig. 4 to Fig. 8, no critical findings or conclusions were driven by the author and the information presented in these time series figures are obviously not clear. I suggest the author consider moving them into supplementary material or giving more discussion.

**A2:** We also considered what reviewer said. While the time series figures in our study are basic data, and are used to be basis of the following calculations. For instance, Fig. 2 and Fig. 3 mainly give the observed results of some variates. Fig. 4 and Fig. 5 are the key parameters in our model, and their time series are the basis of the calculation of extinction coefficients. Fig.6 to Fig.9 are the calculated results that calculated by our model.

For the correlations that reviewer mentioned, the correlation among the visibility, PM2.5 and the total particle number concentration are not the main content. We therefore only give the correlation coefficient.

**Q3:**

3)  I am not satisfied with the author's response to Question 6 from Referee 1 (see acp-2016-119-AC1-supplement-2). The internally mixed and externally mixed results are not shown in Fig. 1, which is not consistent with its figure caption and figure legend or where are the green and red curves?

**A3:** We are so sorry that referee is not satisfied with our reply to Question 6. The Fig.1 we give including the internally mixed and externally mixed results, and found them to be consistent with those of the core-shell model. Maybe the coincidence of lines caused the misunderstanding. Thus, we give them respectively. The values of the extinction coefficient calculated by the core-shell model are between those of the internally mixed model and externally mixed model. The relationships between the model-derived extinction coefficients and measurement-derived extinction coefficients using the three mixed models are given below ($\lambda=550nm$).The result of the internally /externally mixed model is consistent with the current values. Considering the article length and structure, we discussed only the core-shell model in this paper.

[Figure]

Fig.1 Time series of measurement-derived extinction coefficients and model-derived extinction coefficients (core-shell/internally/externally model)

[Figure]

Fig.2 Correlation coefficient of model-derived extinction coefficients (core-shell/internally/externally model) and measurement-derived extinction coefficients

[revised manuscript text omitted]

---

## Author Response (AR3)

Thanks a lot for nice suggestion.

For the grammar and language problems, we have corrected in the paper and marked them out.

For the other problems, we made point to point answers here:

1. Page 2, line 5: maybe three significant figures is too precise.

Yes, we already corrected into 2.4km.

2. Page 5, line 11: 850 professional IC is the name of instrument?

It is the model of the instrument. We already corrected it in the paper into "a chromatograph (Metrohm model 850 professional IC, Switzerland)"

3. Page 5, line 15: Please fix all the units (remove center dot)

Thank you. We already fixed all the units.

4. Page 5, line 31: move this sentence to the end of the previous paragraph.

Yes, we already moved.

5. Page 6, line 11: explain the metrics here, please.

Sorry, what we wanted to express was that the two models had the similar accuracy. We already corrected now into "We used the ion pairing method from Gysel et al. (2007), and his method is as accurate as the ADDEM model (Topping et al., 2005)."

6. Page 8, line 9: Remove this sentence? Relevance?

Yes, we already removed.

7. Page 8, line 12: For ambient measurements, RH is never zero. Did you consider deliquescence?

" RH=0" means the aerosol particles do not uptake water. It is an ideal condition, and reflects optical features of dry particles. We are sorry that we did not express well. We already changed into "If RH is close to 0, then GF=1."

We did not consider deliquescence. In fact, the deliquescence of different aerosols is different because of different composition for the particles. In this study, the hygroscopic increase curves obey the equation 2.

8. Page 8, line 14: Please move this equation after the sentence "... measured by LPS."

We already moved it.

9. Page 9, line 23: This is not the reason but a cause. For different lambda, different aerosol

size ranges are important. This is what you describe below.

We corrected into "because that 940 nm is similar with the light source wavelength of visibility meter."

10. Page 10, line 18: correct unit

Yes, we already correct it into "μm".

11. Page 11, line 29: please reformulate, the figure cannot be divided, but wet extinction coefficients.

Yes, we already corrected into "extinction coefficients shown in fig.9(a) was divided by $b_{ext}$ to produce Fig.10(b)."

12. Page 12, line 4: The smaller sizes coincide with the Mie Maximum.

Page 12, line 8: The aerosol grows due to hygroscopicity, which leads to changes in scale parameter. Different sizes are then at the Mie maximum

I am not sure if I understand the meaning of the expert, what we want to express in the paragraph is that:

[revised manuscript text omitted]